# The interplay of RNA:DNA hybrid structure and G-quadruplexes determines the outcome of R-loop-replisome collisions

Charanya Kumar[1], Sahil Batra[1], Jack D Griffith[2], Dirk Remus[1]*

[1]Molecular Biology Program, Memorial Sloan Kettering Cancer Center, New York, United States; [2]Lineberger Comprehensive Cancer Center and Departments of Microbiology and Immunology, and Biochemistry and Biophysics, University of North Carolina at Chapel Hill, Chapel Hill, United States

**ABSTRACT** R-loops are a major source of genome instability associated with transcription-induced replication stress. However, how R-loops inherently impact replication fork progression is not understood. Here, we characterize R-loop-replisome collisions using a fully reconstituted eukaryotic DNA replication system. We find that RNA:DNA hybrids and G-quadruplexes at both co-directional and head-on R-loops can impact fork progression by inducing fork stalling, uncoupling of leading strand synthesis from replisome progression, and nascent strand gaps. RNase H1 and Pif1 suppress replication defects by resolving RNA:DNA hybrids and G-quadruplexes, respectively. We also identify an intrinsic capacity of replisomes to maintain fork progression at certain R-loops by unwinding RNA:DNA hybrids, repriming leading strand synthesis downstream of G-quadruplexes, or utilizing R-loop transcripts to prime leading strand restart during co-directional R-loop-replisome collisions. Collectively, the data demonstrates that the outcome of R-loop-replisome collisions is modulated by R-loop structure, providing a mechanistic basis for the distinction of deleterious from non-deleterious R-loops.

*For correspondence:
remusd@mskcc.org

Competing interest: The authors declare that no competing interests exist.

## Introduction

Genome maintenance is dependent on the complete and accurate replication of the chromosomal DNA prior to cell division. However, chromosomes present diverse obstacles to normal replication fork progression, including protein-DNA complexes, DNA damage, and non-B-form DNA secondary structures. Among these, R-loops have emerged as critical determinants of transcription-replication conflict (TRC) linked to genome instability in human developmental disorders and disease (*Crossley et al., 2019*; *Garcia-Muse and Aguilera, 2019*). How R-loops inherently impact fork progression is not understood.

R-loops are three-stranded nucleic acid structures that are formed co-transcriptionally when nascent RNA anneals to the DNA template, displacing the non-template strand as single-stranded DNA (ssDNA). Some R-loops serve important physiological roles by promoting chromosome segregation (*Kabeche et al., 2018*), transcription termination (*Skourti-Stathaki et al., 2011*), Ig class switch recombination (*Yu et al., 2003*), telomere homeostasis (*Graf et al., 2017*), or gene expression control (*Niehrs and Luke, 2020*). In contrast, unscheduled R-loops can induce DNA breaks that threaten genome stability. R-loop-induced DNA damage occurs predominantly in S phase, consistent with R-loops exacerbating TRC (*Gan et al., 2011*; *Gomez-Gonzalez et al., 2011*; *Stork et al., 2016*; *Tuduri et al., 2009*; *Wellinger et al., 2006*). However, for reasons that are unknown, only a subset of

unscheduled R-loops appears to cause DNA damage (*Costantino and Koshland, 2018*; *Promonet et al., 2020*). In part, this may be due to head-on (HO) TRC being more detrimental than co-directional (CD) TRC. Yet, collisions in either orientation induce a DNA damage response in human cells (*Hamperl et al., 2017*). This raises the question: what distinguishes harmful from harmless R-loops?

R-loops are detected at 5 % and 8 % of the human and yeast genomes, respectively (*Sanz et al., 2016*; *Wahba et al., 2016*). However, under normal conditions R-loops form transiently, limiting their potential to interfere with DNA replication (*Sanz et al., 2016*; *Wahba et al., 2011*). This dynamism is a consequence of the action of R-loop resolving enzymes, such as RNase H that degrades RNA in RNA:DNA hybrids and various helicases that may unwind RNA:DNA hybrids. Consequently, while R-loops also form under normal conditions (*Chan et al., 2014*; *El Hage et al., 2014*; *Ginno et al., 2012*), mutations in R-loop resolving enzymes dramatically increase cellular R-loop levels and aggravate genome instability (*Crossley et al., 2019*; *Garcia-Muse and Aguilera, 2019*). In addition, co-transcriptional RNA processing and export factors, and topoisomerases, limit R-loop formation (*Huertas and Aguilera, 2003*; *Li and Manley, 2005*; *Tuduri et al., 2009*). Thus, cells employ a host of strategies to protect the genome from deleterious R-loops.

Cellular R-loops range in size from hundreds to thousands of base pairs (*Garcia-Pichardo et al., 2017*; *Malig et al., 2020*; *Wahba et al., 2016*). In general, R-loop formation correlates with high gene activity, GC richness, and G/C skew. The prevalence of G/C skew at R-loops is partly explained by the enhanced stability of RNA:DNA hybrids featuring a G-rich RNA strand (*Roberts and Crothers, 1992*). In addition, stretches of G on the displaced non-template strand promote the formation of G-quadruplexes (G4s) that stabilize R-loops (*De Magis et al., 2019*; *Duquette et al., 2004*). G4s are DNA secondary structures formed by stacks of G-quartets in which four guanine bases form Hoogsteen base pairs in a planar ring configuration (*Burge et al., 2006*). G4 sequences can interfere with DNA replication, causing genetic or epigenetic instability (*Paeschke et al., 2013*; *Ribeyre et al., 2009*; *Sarkies et al., 2010*). The mechanisms involved are not clear, but G4 sequences on either the leading or the lagging strand template have been reported to interfere with normal DNA replication (*Dahan et al., 2018*; *Lopes et al., 2011*; *Sarkies et al., 2010*). As for RNA:DNA hybrids, a host of helicases has been implicated in promoting DNA replication through G4 sequences (*Lerner and Sale, 2019*). How G4s may modulate the impact of R-loops on DNA replication is not known.

Cellular studies are limited in their ability to differentiate direct and indirect effects of R-loops. The latter may include enhanced RNAP stalling (*Belotserkovskii et al., 2010*), establishment of repressive chromatin structures (*Castellano-Pozo et al., 2013*; *Garcia-Pichardo et al., 2017*; *Garcia-Rubio et al., 2018*), or induction of DNA breaks by nucleases (*Sollier et al., 2014*; *Su and Freudenreich, 2017*). To overcome this limitation, here, we employ the reconstituted budding yeast DNA replication system in combination with purified R-loop templates to study orientation-specific R-loop-replisome collisions (*Devbhandari et al., 2017*; *Devbhandari and Remus, 2020*; *Yeeles et al., 2015*). We demonstrate that both CD and HO R-loops can adversely affect fork progression, dependent on the specific configuration of RNA:DNA hybrids and G4s. RNA:DNA hybrid- and G4-induced defects can be resolved by RNase H1 and Pif1, respectively. Unexpectedly, we find that leading strand synthesis can be reprimed by distinct mechanisms downstream of G4s and RNA:DNA hybrids, promoting continued fork progression at R-loops. Collectively, our data reveals how the specific structure of R-loops determines the outcome of R-loop-dependent TRC.

## Results

### Preparation and characterization of R-loop-containing DNA templates

To generate DNA templates harboring R-loops of characteristic length and nucleotide composition, we inserted a 1.4 kbp fragment derived from the R-loop-forming *Airn* locus under control of a T7 promoter into yeast replication origin-containing plasmids (*Figure 1A*; *Carrasco-Salas et al., 2019*; *Ginno et al., 2012*). Studies in human cells have demonstrated that R-loops formed at this sequence impede normal fork progression (*Hamperl et al., 2017*). Both the template and non-template strand harbor several G4 sequences, but G4 potential is greater on the non-template strand (*Figure 1A* and *Figure 1—figure supplement 1A*). To reconstitute orientation-specific collisions of replisomes with R-loops, we inserted the R-loop element in both orientations relative to the replication origin, *ARS305*. In either orientation, the C-rich strand serves as the template for

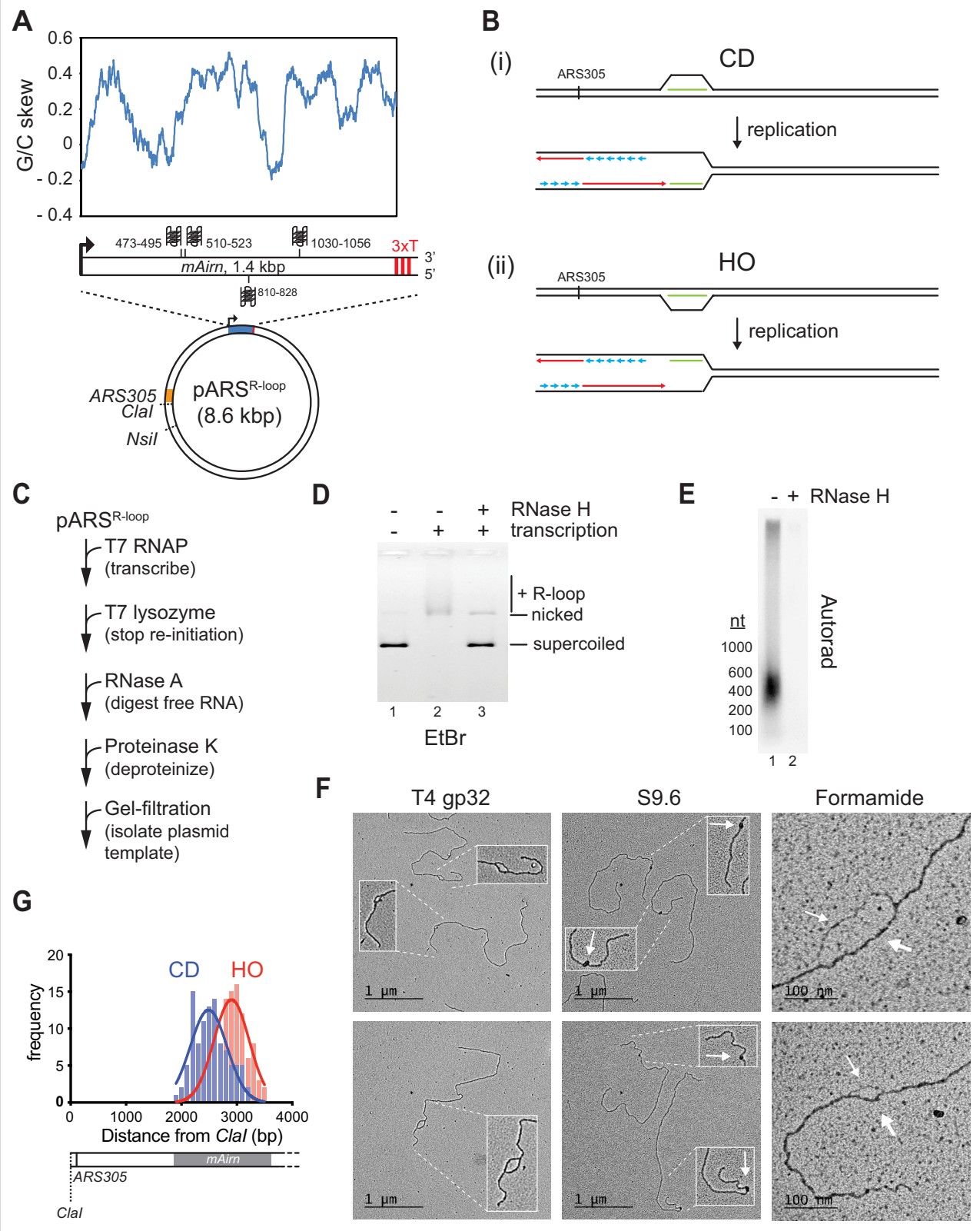

**Figure 1.** Preparation and characterization of R-loop-containing templates. (**A**) Schematic of R-loop plasmid template. Plot shows G/C skew at *Airn* sequence. Graphic shows positions of potential G-quadruplexes composed of stacks of three G-quartets in non-template (top) and template (bottom) strand. 3 × T: T7 terminator tandem repeat. (**B**) Schematic of co-directional (CD) and head-on (HO) R-loop-replisome collisions in experimental setup. Template strands: black; leading strand: red; lagging strand: blue; RNA: green. (**C**) Reaction scheme for preparation of R-loop-containing template. (**D**)

*Figure 1 continued on next page*

*Figure 1 continued*

Native agarose gel analysis of purified plasmid template. The gel was stained with ethidium bromide. (**E**) R-loop-containing template harboring $^{32}$P-labeled RNA was mock-treated or digested with RNase H and analyzed by denaturing formaldehyde agarose gel-electrophoresis and autoradiography. (**F**) Electron microscopy (EM) analysis of R-loop templates. White arrows in center panels indicate S9.6-specific density. Thin and thick arrows in right panels indicate displaced non-template or RNA:DNA duplex, respectively. (**G**) Frequency distribution of R-loop distances from *ClaI* site in CD and HO orientation.

The online version of this article includes the following figure supplement(s) for figure 1:

**Source data 1.** Preparation and characterization of R-loop-containing templates.

**Source data 2.** Preparation and characterization of R-loop-containing templates.

**Source data 3.** Preparation and characterization of R-loop-containing templates.

**Source data 4.** Preparation and characterization of R-loop-containing templates.

**Source data 5.** Preparation and characterization of R-loop-containing templates.

**Figure supplement 1.** R-loop template preparation and characterization by atomic force microscopy (AFM).

**Figure supplement 1—source data 1.** R-loop template preparation and characterization by atomic force microscopy (AFM).

**Figure supplement 1—source data 2.** R-loop template preparation and characterization by atomic force microscopy (AFM).

transcription. Linearization of the plasmid template ensures that only one of the two replication forks emanating from the origin encounters the R-loop, while the other fork runs off the opposite template end (*Figure 1B*).

To promote R-loop formation in vitro, transcription is carried out on negatively supercoiled plasmid templates (*Drolet et al., 1994*; *Stolz et al., 2019*). Moreover, to avoid RNA discontinuities due to repeated transcription initiation, purified T7 lysozyme, which inhibits transcription initiation but not elongation by T7 RNAP (*Kumar and Patel, 1997*; *Zhang and Studier, 1997*), is added late in transcription reactions (*Figure 1C*). Subsequently, RNase A is used to digest free RNA, which facilitates the separation of the template from excess free RNA by size-exclusion chromatography (*Figure 1—figure supplement 1B+C*). Reaction mixtures are deproteinized to remove T7 RNAP and templates are purified by gel filtration (*Figure 1—figure supplement 1C*). R-loop formation under these conditions is efficient as demonstrated by the RNase H-sensitive gel-electrophoretic mobility shift of the template DNA (*Figure 1D*).

Analysis of the R-loop-associated RNA by denaturing formaldehyde agarose gel-electrophoresis reveals that most RNA molecules are significantly shorter than 1.4 kb, ranging in size between ~200 and 600 nt (*Figure 1E*). Thus, only sections of the *Airn* sequence form R-loops in vitro, which is consistent with previous single-molecule R-loop sequencing and footprinting data (*Carrasco-Salas et al., 2019*). To characterize the R-loop templates, we employed electron microscopy (EM) after rotary shadow casting (*Figure 1F*). In this approach, R-loop positions are mapped relative to the template ends. We employed three approaches to image R-loops by EM. To visualize the displaced non-template strand, DNA templates were incubated with ssDNA binding protein, T4 gp32. This approach reveals bubble-like R-loop structures of variable sizes specifically at the *Airn* sequence. Relevant to the analyses below, R-loops form preferentially in the 5' half of the transcribed sequence, resulting in their center of distribution being closer to the replication origin in the CD than in the HO orientation (*Figure 1G*). To probe for RNA:DNA hybrids, templates were incubated with the RNA:DNA hybrid-specific antibody, S9.6 (*Figure 1F*; *Boguslawski et al., 1986*). This analysis reveals the formation of unique electron-dense structures at the position of the *Airn* sequence, confirming the presence of RNA:DNA hybrids at this site (*Figure 1—figure supplement 1D*). Finally, DNA templates were spread on EM grids under mildly denaturing conditions in the presence of formamide (*Figure 1F*). This analysis reveals a characteristic difference in the thickness of the two arms of the R-loops, consistent with one arm corresponding to the single-stranded non-template strand and the other corresponding to the RNA:DNA duplex. As in the analysis with T4 gp32, and consistent with the gel analysis in *Figure 1E*, R-loops observed in the presence of formamide exhibit a range of sizes, demonstrating that the size variations are not due to the sample preparation method. This is further confirmed by similar observations using atomic force microscopy (AFM) (*Figure 1—figure supplement 1E*). In conclusion, the data demonstrates efficient formation of canonical R-loop structures of variable size and position at the *Airn* sequence in vitro.

## Both CD and HO R-loops perturb normal fork progression

To test the impact of R-loops on fork progression, DNA templates are linearized with *NsiI* prior to origin firing. Replication products are digested with *ClaI* prior to gel analysis to reduce heterogeneity in their gel mobility caused by the distributive initiation of strand synthesis at the origin (*Figure 2A+B*; *Aria and Yeeles, 2018*). In the absence of R-loops, the *Airn* sequence does not present a notable obstacle to fork progression in either orientation. In contrast, CD or HO R-loops induce a striking loss of full-length replication products and the appearance of new replication intermediates. Native gel analysis reveals reduced levels of full-length linear daughter molecules in the presence of either CD or HO R-loops, as well as the appearance of stalled forks and partially replicated daughter molecules resulting from the uncoupling of DNA unwinding from leading strand synthesis (*Figure 2C, D*). Consistent with the native gel data, nascent strand analysis on denaturing gels demonstrates that CD and HO R-loops cause a decrease in full-length rightward leading strands by ~75 % and ~55%, respectively (*Figure 2C*). The levels of leftward leading strands are not affected, confirming that the loss of full-length rightward leading strands is not due to reduced origin activity. Instead, the loss of full-length rightward leading strands correlates with the generation of prominent ~2.3  or ~2.8 kb stall products at CD and HO R-loops, respectively. The heterogeneity of the stalled leading strand products is reduced after *ClaI* digest, confirming that they originate near *ARS305* (*Figure 2—figure supplement 1A*). Accordingly, leading strand stalling coincides with the *Airn* sequence, which spans the region 1.8–3.2 kb downstream of the origin (*Figure 2A*). The difference in the position of the leading strand stall sites at CD and HO R-loops is, therefore, attributable to the asymmetric distribution of R-loops in the *Airn* sequence (*Figure 1G*). The heterogeneity in stall products remaining after *ClaI* digest is likely a consequence of variations in R-loop sizes and positions (*Figure 1*).

Two-dimensional gel analysis demonstrates that leading strand stalling at CD and HO R-loops results from both fork stalling and helicase uncoupling (*Figure 2E*). Time course analyses demonstrate that leading strands stall within minutes of origin firing and remain stable for at least 2 hr (*Figure 2—figure supplement 1B*, *Figure 3*). Similarly, native gel analysis reveals that a fraction of forks remains stably stalled specifically in the presence of R-loops. Fork uncoupling occurs after leading strand stalling, as expected due to the reduced rate of DNA unwinding by CMG upon uncoupling from leading strand synthesis (*Devbhandari and Remus, 2020*; *Taylor and Yeeles, 2019*).

In addition to stalled leading strands, denaturing gel analysis reveals the formation of novel ~5.5  and ~5.0 kb products indicative of leading strand restart downstream of CD and HO R-loops, respectively (*Figure 2C+D*). The identity of these products is confirmed by multiple lines of evidence: First, these products are sensitive to cleavage by the leading strand-specific nicking enzyme *Nb.BbvCI*, but not lagging strand-specific *Nt.BbvCI*, demonstrating that they are nascent leading strand products (*Figure 2—figure supplement 2A*). Second, these leading strand products are not sensitive to *ClaI* cleavage, indicating that they do not originate near *ARS305* (*Figure 2—figure supplement 1A*). Third, in regular time course experiments, these products accumulate with slower kinetics than stalled leading strands, as expected for restart following leading strand stalling (*Figure 2—figure supplement 1B*). Fourth, in experiments in which nascent strands are pulse-labeled with α[$^{32}$P]-dATP for 2.5 min following origin activation before being chased with excess cold dATP restart products are not detectable, as expected if they form after origin activation (*Figure 3*). Moreover, below we will show that this leading strand restart occurs at G4s in the leading strand template.

We conclude that both CD and HO R-loops can perturb replication fork progression, which is consistent with observations in human cells (*Hamperl et al., 2017*). Replication abnormalities in either orientation include fork stalling, uncoupling of leading strand synthesis from fork progression, and discontinuous leading strand synthesis involving leading strand restart. A fraction of R-loops in either orientation is also bypassed by replisomes without disruption. This diversity in outcomes is explained by the heterogeneity of R-loops in our templates (*Figure 1*). Because the replicative DNA helicase, CMG, can efficiently bypass steric blocks on the displaced lagging strand (*Fu et al., 2011*; *Kose et al., 2019*), fork stalling is likely induced by obstacles on the leading strand. This raises questions about the fork-stalling mechanism as forks will encounter either RNA:DNA hybrids or the displaced non-template strand on the leading strand during CD and HO R-loop-replisome collisions, respectively. We, therefore, investigated the molecular basis for replication aberrations at R-loops.

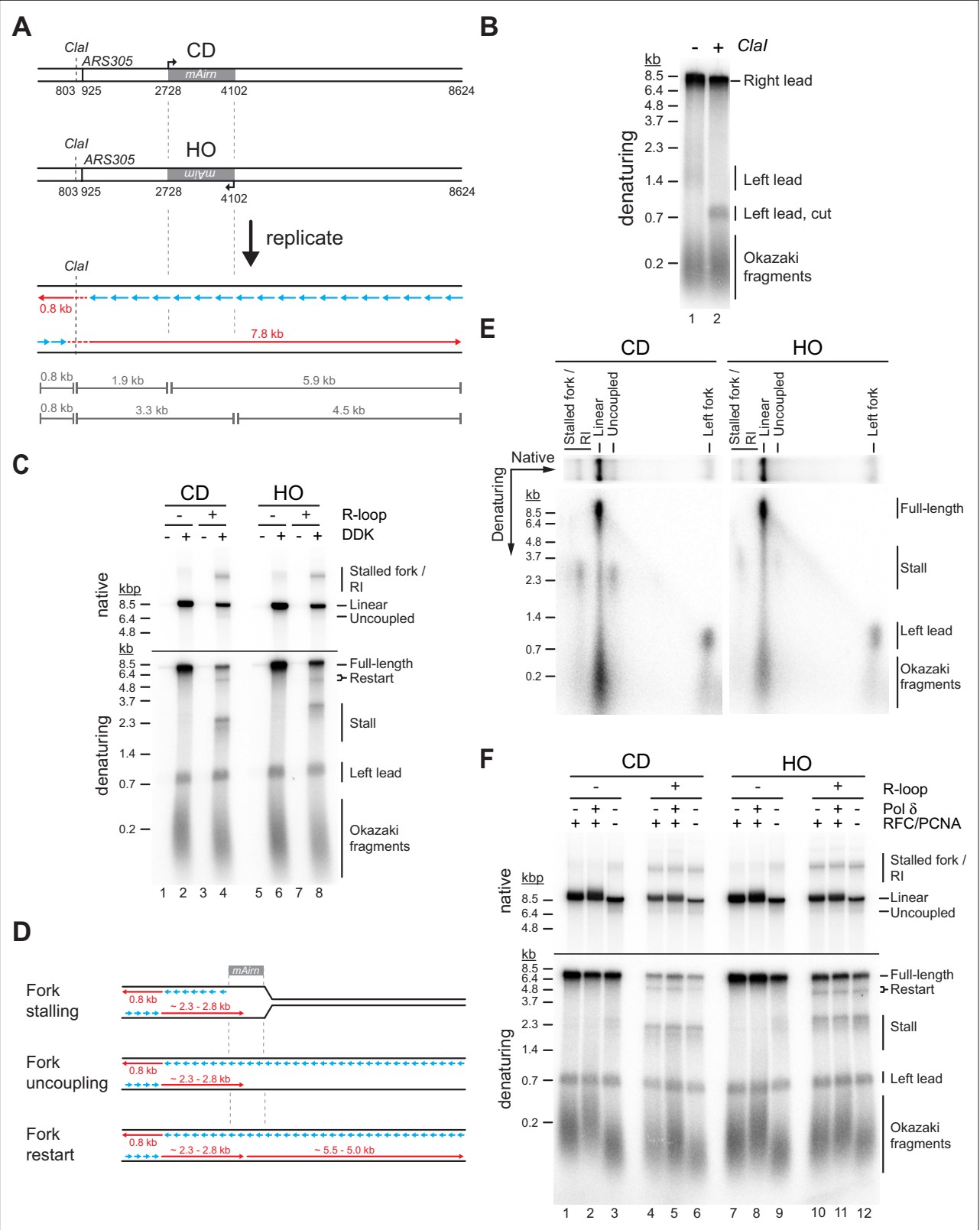

**Figure 2.** Both co-directional (CD) and head-on (HO) R-loops perturb normal fork progression. (**A**) Schematic illustrating expected sizes of replication products. (**B**) Denaturing agarose gel analysis of replication products obtained on R-loop-free template. Left lead: Leftward leading strands; Right lead: Rightward leading strands. (**C**) Native (top) and denaturing (bottom) agarose gel analyses of replication products obtained on templates harboring *Airn* sequence in CD or HO orientation. Stall: Stalled rightward leading strands; Restart: Rightward leading strand restart product; Full-length: full-length

*Figure 2 continued on next page*

*Figure 2 continued*

rightward leading strand; RI: replication intermediates. (**D**) Schematic illustrating replication products observed in (**C**). (**E**) Two-dimensional gel analysis of replication products obtained in presence of R-loops (corresponding to lanes 4 and 8 in (**C**)). Products were digested with *ClaI*. (**F**) Replication products obtained in the absence or presence of RFC/PCNA or Pol δ.

The online version of this article includes the following figure supplement(s) for figure 2:

**Source data 1.** Both co-directional (CD) and head-on (HO) R-loops perturb normal fork progression.

**Source data 2.** Both co-directional (CD) and head-on (HO) R-loops perturb normal fork progression.

**Source data 3.** Both co-directional (CD) and head-on (HO) R-loops perturb normal fork progression.

**Figure supplement 1.** Both co-directional (CD) and head-on (HO) R-loops perturb normal fork progression.

**Figure supplement 1—source data 1.** Characterization of replisome encounters with co-directional (CD) and head-on (HO) R-loops.

**Figure supplement 1—source data 2.** Characterization of replisome encounters with co-directional (CD) and head-on (HO) R-loops.

**Figure supplement 2.** Characterization of replisome encounters with co-directional (CD) and head-on (HO) R-loops.

**Figure supplement 2—source data 1.** Characterization of replisome encounters with co-directional (CD) and head-on (HO) R-loops.

## Fork stalling at R-loops is not dependent on Tof1

Csm3, Tof1, and Mrc1 (CTM) form a fork protection complex that associates with replisomes to promote normal fork progression, prevent the uncoupling of replisomes from DNA synthesis after nucleotide depletion by hydroxyurea, and maintain fork integrity during the replication of structure-forming trinucleotide repeats (*Bando et al., 2009*; *Gellon et al., 2019*; *Katou et al., 2003*; *Szyjka et al., 2005*; *Tourriere et al., 2005*; *Yeeles et al., 2017*). In addition, Tof1 mediates fork pausing at protein-DNA complexes, such as the rDNA replication fork barrier, tRNA genes, and centromeres (*Calzada et al., 2005*; *Hodgson et al., 2007*; *Tourriere et al., 2005*). We find that fork stalling, uncoupling, and restart at CD and HO R-loops are not affected by CTM, demonstrating that the mechanism of fork stalling at R-loops is distinct from that at protein-DNA barriers (*Figure 2—figure supplement 2B*).

## PCNA suppresses helicase-polymerase uncoupling at *Airn* sequence

A previous study found that RNA at 5' primer-template junctions promotes strand displacement by Pol δ (*Stith et al., 2008*). We, therefore, tested if Pol δ may promote fork progression through R-loops (*Figure 2F*). On R-loop-containing templates, replication intermediates obtained in the absence of Pol δ or its processivity factor PCNA (together with its loader RFC) were indistinguishable from those obtained with complete replisomes. Thus, strand displacement by Pol δ does not promote fork progression at R-loops. The data also demonstrates that Pol δ is not required for leading strand restart under these conditions. Intriguingly, however, we note that the absence of PCNA increases leading strand stalling and fork uncoupling at the *Airn* sequence even in the absence of R-loops (*Figure 2F*, lanes 3 + 9), indicating a role for PCNA in maintaining the coupling of replisomes to leading strand synthesis at G4 sequences. We speculate that PCNA mediates this function through stabilization of the leading strand polymerase, Pol ε, on the template.

## RNase H1 promotes fork passage specifically at CD R-loops

RNase H1 overexpression is commonly used to assess the contribution of R-loops to TRC. We, therefore, tested how purified yeast RNase H1 (*Figure 4A*) affects fork progression at R-loops in vitro. At CD R-loops, RNase H1 decreased the levels of stalled and uncoupled replication intermediates while increasing the formation of full-length replication products, suggesting that RNA:DNA hybrids on the leading strand can impede fork progression (*Figure 4B+C*, lanes 1–4; *Figure 4—figure supplement 1A*). In contrast, leading strand restart at CD R-loops was not affected by RNase H1, indicating that restart is not a direct consequence of RNA:DNA hybrids. Instead, this suggests a role for G4s in leading strand restart at CD R-loops (*Figure 4—figure supplement 1B*), which is also supported by data below. While not being dependent on RNA:DNA hybrid persistence, we note that formation of the restart-inducing structures is dependent on transcription (*Figure 4B*, compare lanes 1 + 2–3 + 4). Since G4s on the lagging strand template are unlikely to be involved in leading strand restart (see below, *Figure 4E*), this data suggests that transcription induces G4s not only on the displaced non-template strand, but to some extent also on the template strand, which forms the leading strand

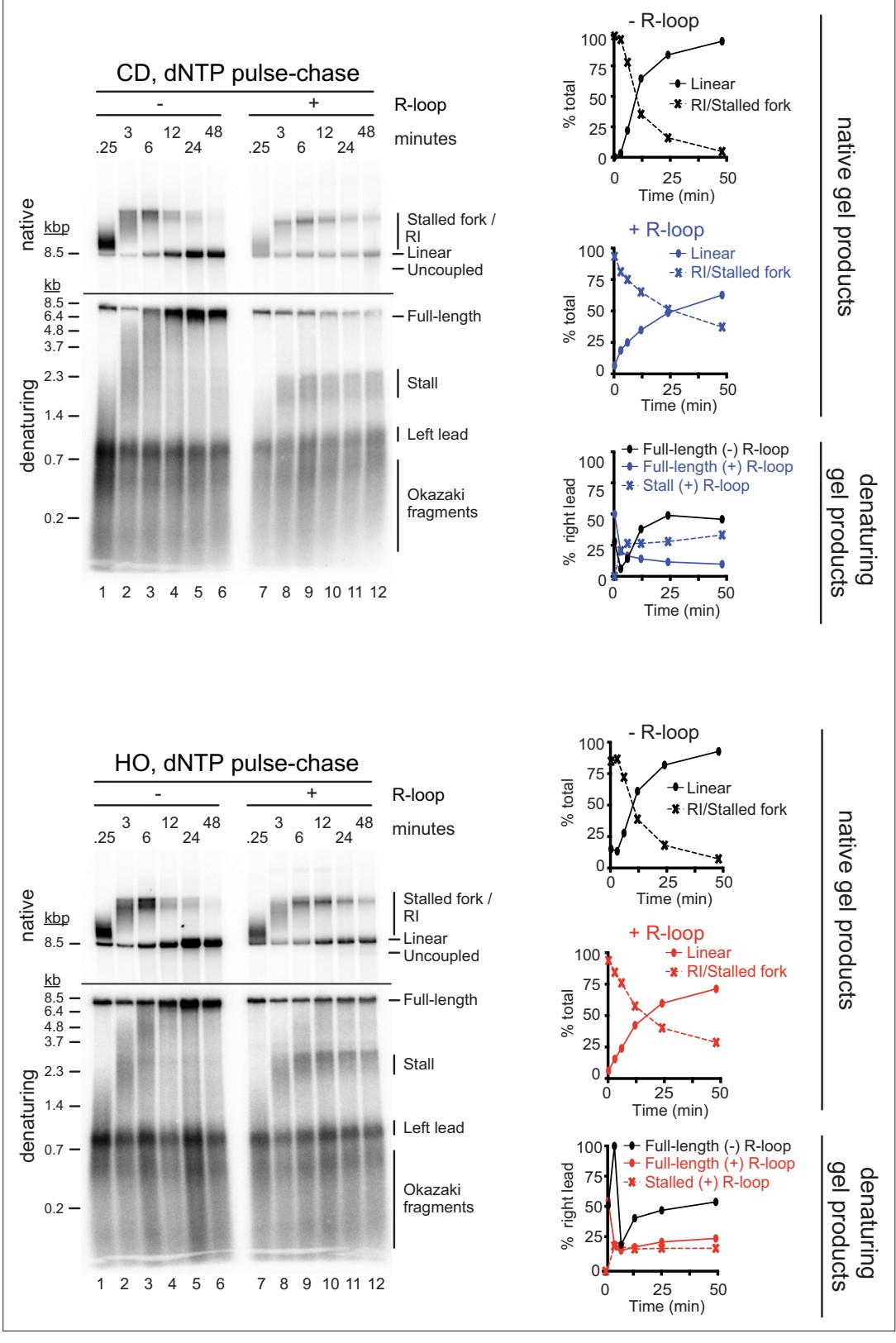

**Figure 3.** dNTP pulse-chase analysis of fork progression through co-directional (CD) and head-on (HO) R-loops. Pulse-chase time course analysis of replication reactions on CD (top) and HO (bottom) templates. Signal intensities of replication products were quantified and plotted as percentage of total signal.

*Figure 3 continued on next page*

*Figure 3 continued*

The online version of this article includes the following figure supplement(s) for figure 3:

**Source data 1.** dNTP pulse-chase analysis of fork progression through co-directional (CD) and head-on (HO) R-loops.

---

template for the replisome at CD R-loops. While it is not clear how transcription induces G4s on the template strand in our system, a plausible mechanism may involve negative supercoiling of G4 sequences upstream of T7 RNAP (*Zhang et al., 2020*).

In contrast, fork stalling and uncoupling were largely unaffected by RNase H1 at HO R-loops, indicating that RNA:DNA hybrids on the lagging strand are not the direct cause for these events (*Figure 4B*, lanes 5–8). Instead, this data again suggests that G4s on the leading strand template, formed by the G-rich displaced strand in this orientation, cause fork stalling or uncoupling. Moreover, the insensitivity of the fork block at HO R-loops to RNase H1 demonstrates that G4s on the leading strand template that have the potential to stall the replication fork, while dependent on transcription for formation, are not dependent on the persistence of RNA:DNA hybrids on the lagging strand template. In contrast, since leading strand restart at HO R-loops is sensitive to RNase H1, G4s that block the leading strand polymerase, but not the CMG, appear to be stable only in ssDNA, that is, in the presence of the RNA:DNA hybrid on the opposite strand. We hypothesize that the heterogeneity in the size and position of R-loops in our R-loop template (*Figure 1*) results in the formation of a sub-population of HO R-loops harboring G4s with fork blocking potential that are stable even after RNase H digestion of the RNA:DNA hybrid (*Figure 4—figure supplement 2A*), while another sub-population of HO R-loops harbors G4s only with specific polymerase blocking potential that are resolved after RNase H digestion of the R-loop (*Figure 4—figure supplement 2B*).

## Pif1 promotes fork passage specifically at HO R-loops

The implication of G4s in the fork block at R-loops led us to examine the effect of the Pif1 helicase on fork progression at R-loops. In vitro studies have demonstrated that Pif1 exhibits an evolutionarily conserved specificity for binding and unwinding G4 DNA (*Byrd and Raney, 2015*; *Duan et al., 2015*; *Hou et al., 2015*; *Paeschke et al., 2013*; *Sanders, 2010*; *Wallgren et al., 2016*; *Zhou et al., 2014*). Moreover, Pif1 has been demonstrated to promote fork progression through G4 sequences in vivo and to suppress genomic instability at such sites (*Dahan et al., 2018*; *Paeschke et al., 2013*; *Paeschke et al., 2011*; *Ribeyre et al., 2009*). In addition, Pif1 exhibits enhanced activity on RNA:DNA hybrids in vitro (*Boule and Zakian, 2007*; *Chib et al., 2016*; *Zhou et al., 2014*).

In contrast to RNase H1, we find that Pif1 does not promote fork progression through CD R-loops, supporting the notion that its limited processivity is insufficient to resolve long RNA:DNA hybrids (*Figure 3C*, lanes 1–3) (*Pohl and Zakian, 2019*). However, RNase H1 and Pif1 together effectively eliminate fork stalling, uncoupling, and restart events CD R-loops (*Figure 4C*, lane 4). This suggests that RNase H1 and Pif1 coordinately promote fork progression at CD R-loops by eliminating both RNA:DNA hybrids and G4s from the leading strand template. Conversely, at HO R-loops Pif1 was markedly more efficient than RNase H1 in promoting fork progression (*Figure 4C*, lanes 5–7), which is consistent with the notion that G4s on the leading strand template cause fork stalling at HO R-loops. Accordingly, addition of RNase H1 did not further promote fork progression at HO R-loops in the presence of Pif1 (*Figure 4C*, lane 8). Together, the data suggests that fork stalling at R-loops can be induced by RNA:DNA hybrids and G4s on the leading strand template, while neither structure presents a fork block when formed on the lagging strand.

## G4s and RNA:DNA hybrids on the leading strand template can pose fork blocks

To directly test the impact of G4s on fork progression, we performed reactions in the presence of the G4-stabilizer, pyridostatin (PDS). On DNA templates that lack the *Airn* sequence, PDS does not impede DNA replication (*Figure 4D*, lanes 1 + 2). In contrast, PDS induces a strong block to fork progression specifically at the *Airn* sequence in the HO orientation, that is, when the G-rich strand forms the leading strand template, even in the absence of R-loops (*Figure 4D*, lanes 3–6). Thus, G4s forming on the G-rich strand of the *Airn* sequence in the absence of transcription impede replication

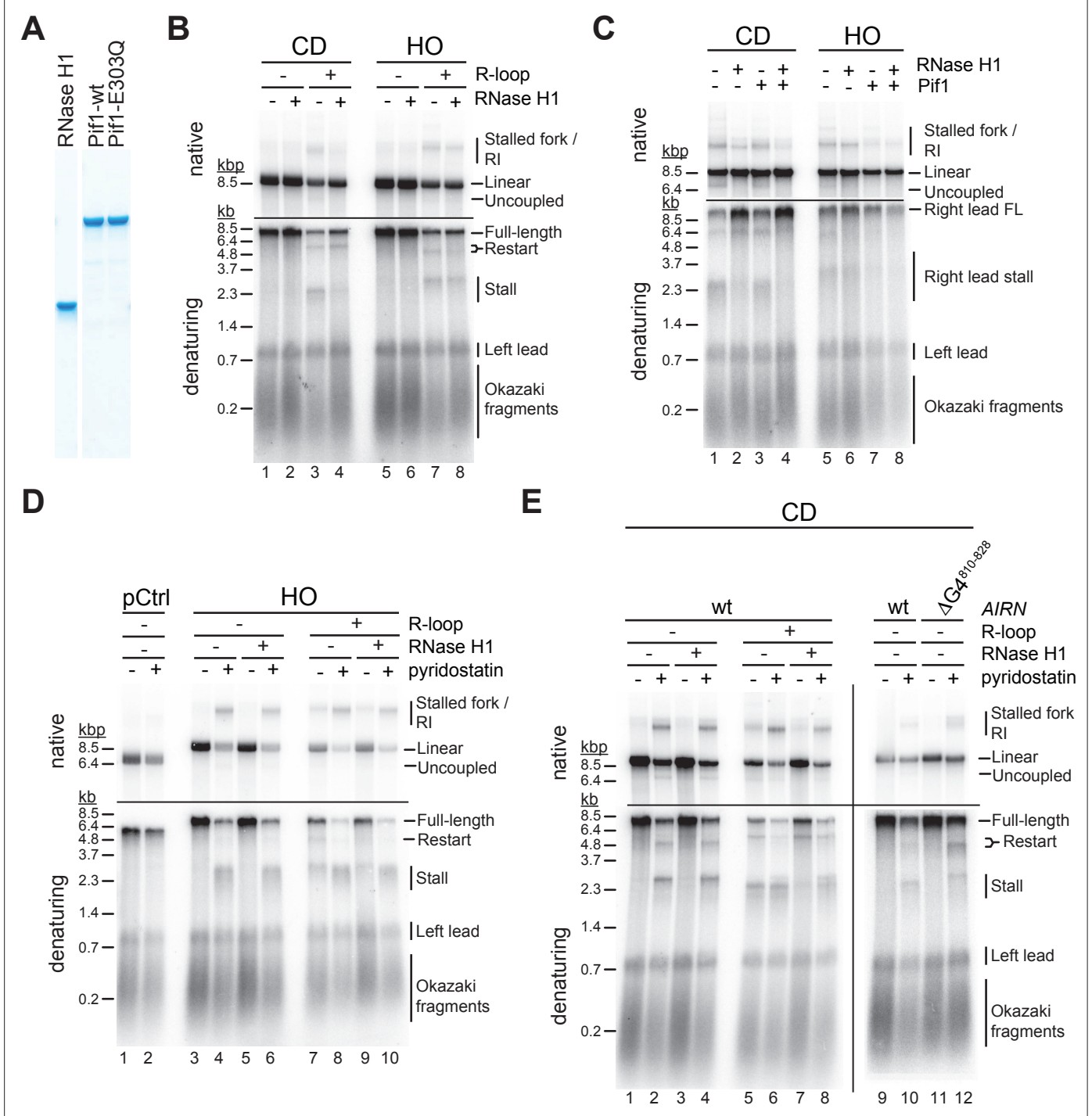

**Figure 4.** G-quadruplexes (G4s) and RNA:DNA hybrids pose impediment to leading strand synthesis that can be resolved by Pif1 or RNase H1, respectively. (**A**) Purified proteins analyzed by SDS-PAGE and Coomassie stain. (**B**) Replication products obtained on co-directional (CD) or head-on (HO) templates in the absence or presence of RNase H1. (**C**) Replication products obtained on CD or HO templates in the absence or presence of RNase H1 and Pif1. (**D**) Replication products obtained on HO templates in the absence or presence of RNase H1 and pyridostatin. (**E**) Replication products obtained on CD templates in the absence or presence of RNase H1 and pyridostatin. ΔG4$^{810-828}$: *Airn* sequence containing deletion of G4 sequence at position 810–828.

The online version of this article includes the following figure supplement(s) for figure 4:

**Source data 1.** G-quadruplexes (G4s) and RNA:DNA hybrids pose impediment to leading strand synthesis that can be resolved by Pif1 or RNase H1, respectively.

*Figure 4 continued on next page*

*Figure 4 continued*

**Source data 2.** G-quadruplexes (G4s) and RNA:DNA hybrids pose impediment to leading strand synthesis that can be resolved by Pif1 or RNase H1, respectively.

**Source data 3.** G-quadruplexes (G4s) and RNA:DNA hybrids pose impediment to leading strand synthesis that can be resolved by Pif1 or RNase H1, respectively.

**Figure supplement 1.** RNase H1 promotes fork passage specifically at co-directional (CD) R-loops.

**Figure supplement 2.** RNase H1 promotes fork passage specifically at co-directional (CD) R-loops.

forks only when stabilized by PDS (*Figure 4D*, lanes 3 + 5), demonstrating that G4 stability determines the fork block potential of G4s. In contrast, fork stalling is induced at HO R-loops even in the absence of PDS, and this effect is enhanced by PDS (*Figure 4D*, lanes 7–10). Thus, R-loop formation modulates the G4 composition on the displaced strand, which is consistent with studies demonstrating a cooperative relationship between R-loops and G4s (*De Magis et al., 2019*; *Duquette et al., 2004*; *Lee et al., 2020*). Since RNase H1 is unable to prevent fork stalling at HO R-loops, R-loop formation induces the formation of fork-stalling G4s on the displaced strand but is not required for their maintenance.

In the CD orientation, PDS also induces fork stalling at the *Airn* sequence in the absence of R-loops, but the extent of stalling is less pronounced than in the HO orientation (*Figure 4E*, lanes 1–4; *Figure 1—figure supplement 1A*). In addition, in the absence of R-loops, PDS induces leading strand restart at the *Airn* sequence, demonstrating that replisomes can pass some PDS-stabilized G4s and reprime leading strand synthesis downstream. Deletion of G4$^{810\text{-}828}$ (*Figure 1A*; *Figure 1—figure supplement 1A*) eliminates the major PDS-induced fork block in the CD orientation, identifying this G4 as a fork block (*Figure 4E*, lanes 9–12). This data demonstrates that PDS-induced fork stalling in the CD orientation is caused by G4s in the C-rich leading strand template and not by G4s in the G-rich lagging strand template. The deletion of G4$^{810\text{-}828}$ exposes a novel downstream leading strand stall site that correlates with greatly increased leading strand restart in the presence of PDS (*Figure 4E*, lanes 11 + 12). We conclude that G4s can induce fork stalling or uncoupling depending on G4 stability, which is consistent with the correlation of genetic instability with G4 stability in vivo (*Piazza et al., 2015*).

Replisome uncoupling from leading strand synthesis at PDS-stabilized G4s indicates that some G4s pose a block to the leading strand polymerase, Pol ε, but not the CMG helicase (*Figure 4E*, native gel, lanes 2 + 4). Importantly, while a fraction of uncoupling events at G4s leads to persistent unwinding in the absence of DNA synthesis, we also observe significant leading strand restart downstream of G4s. We note that the leading strand restart efficiency observed at G4s appears to be greater than that observed previously at leading strand DNA damage (*Taylor and Yeeles, 2018*), which we will discuss further below. Strikingly, unlike PDS-induced fork stall events in the CD orientation, CD R-loop-induced fork stall events are sensitive to RNase H1, occur at a position closer to the origin-proximal side of the *Airn* sequence, and are not exacerbated by PDS (*Figure 4E*, compare lanes 5 + 6–1 + 2). This indicates that fork stalling at R-loops in the CD orientation is caused by RNA:DNA hybrids on the leading strand template. Accordingly, stall sites are shifted downstream at CD R-loops in the presence of both RNase H1 and PDS (*Figure 4E*, lanes 6 + 8).

In summary, the data demonstrates that both RNA:DNA hybrids and G4s on the leading strand can induce fork stalling during R-loop-replisome collisions. Moreover, both RNA:DNA hybrids and G4s have the potential to induce uncoupling of leading strand synthesis from fork progression. At G4s transient uncoupling of leading strand synthesis from fork progression is frequently followed by leading strand restart. Notably, a fraction of R-loops in either orientation is also bypassed by forks without disruption. This diversity in outcomes is likely a consequence of the structural heterogeneity of R-loops.

## CMG can unwind or translocate on RNA:DNA hybrids

To determine the molecular basis for the diversity in outcomes of R-loop-replisome collisions, we examined the helicase activity of purified CMG (*Figure 5A*) on RNA:DNA hybrid- and G4-containing substrates. During DNA unwinding at replication forks, CMG translocates on the single-stranded leading strand template in 3′–5′ direction while sterically displacing the lagging strand template. Additionally, purified yeast and human CMG have been demonstrated to be able to translocate on

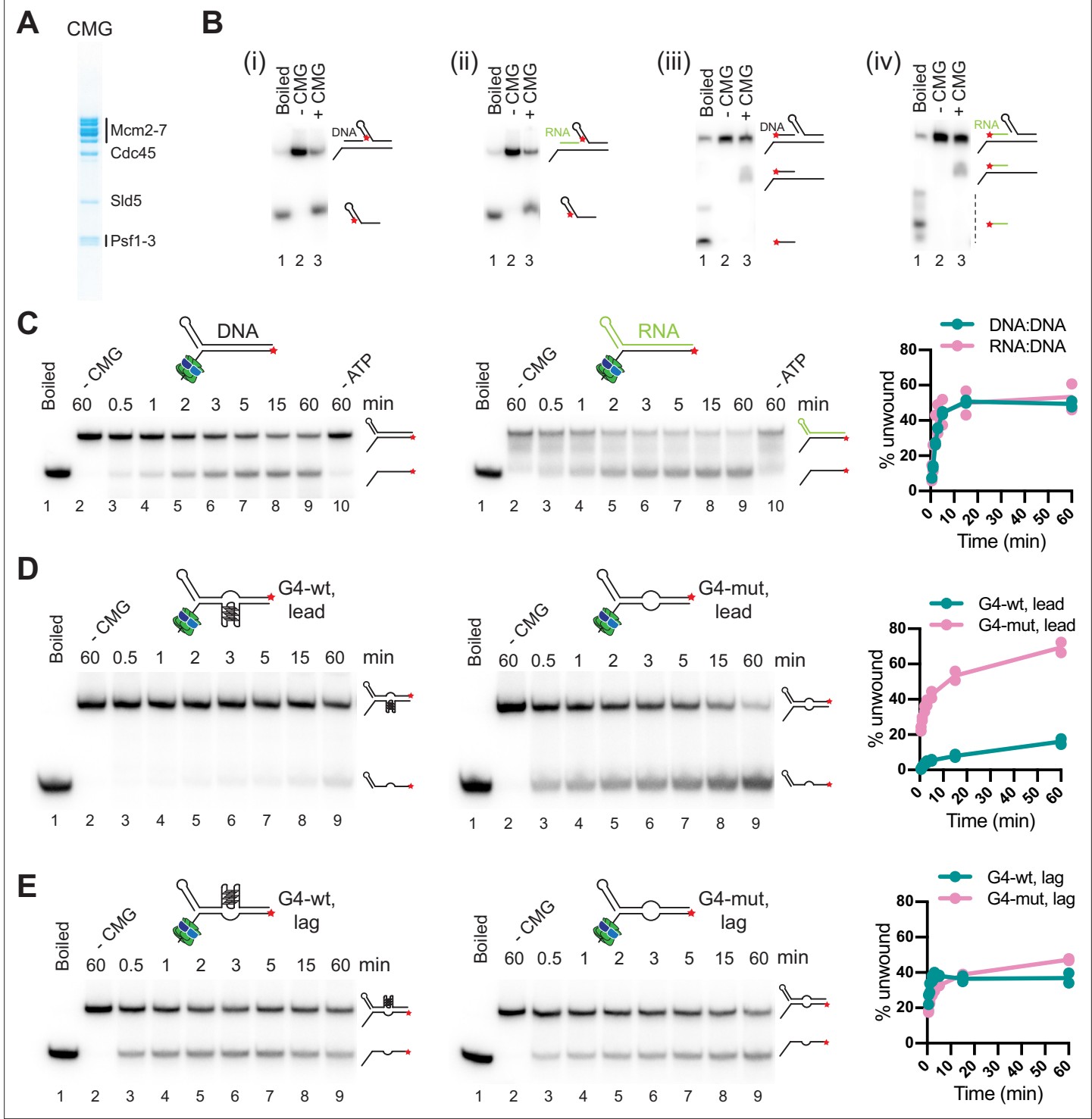

**Figure 5.** CMG can unwind or translocate on RNA:DNA hybrids, while G-quadruplexes (G4s) can block DNA unwinding by CMG. (**A**) Purified CMG. (**B**) Helicase assays with 40 bp forked DNA duplex preceded by 40 bp DNA (i + iii) or RNA:DNA (ii + iv) duplex. ★ indicates position of 5'-$^{32}$ P label. Products were analyzed by native PAGE and autoradiography. (**C**) CMG helicase activity on 60 bp forked DNA (left) or RNA:DNA duplex (right). Plot shows average of two replicates. (**D**) CMG helicase activity on 60 bp substrate harboring wildtype (left) or mutant (right) G4 sequence on the template strand ('lead'). (**E**) As (**D**), with wildtype (left) or mutant (right) G4 sequence on the non-template strand ('lag').

The online version of this article includes the following figure supplement(s) for figure 5:

**Source data 1.** CMG can unwind or translocate on RNA:DNA hybrids, while G-quadruplexes (G4s) can block DNA unwinding by CMG.

**Source data 2.** CMG can unwind or translocate on RNA:DNA hybrids, while G-quadruplexes (G4s) can block DNA unwinding by CMG.

duplex DNA, which is supported by studies in *Xenopus* suggesting that CMG translocates on double-stranded DNA (dsDNA) after replication termination or upon encounter of nicks in the lagging strand template (*Kang et al., 2012*; *Langston and O'Donnell, 2017*; *Low et al., 2020*; *Vrtis et al., 2021*). To determine if CMG can translocate on RNA:DNA hybrids, we constructed oligonucleotide substrates composed of a 40 bp forked duplex DNA preceded by a 40 bp DNA or RNA:DNA duplex that forms a flush 5' ss/dsDNA junction downstream of a 40 nt poly(dT) 3' tail, which serves as the loading site for CMG. A 5' hairpin-like secondary structure promotes the steric unwinding of the downstream oligonucleotide by an approaching CMG (*Kose et al., 2019*). Using this approach, we find that the downstream oligonucleotide is unwound with similar efficiencies by CMG if preceded by a DNA or RNA:DNA duplex (*Figure 5B* i + ii). Importantly, unwinding of the downstream oligonucleotide is not accompanied by the unwinding of the upstream DNA or RNA oligonucleotide (*Figure 5B* iii + iv). Thus, CMG can translocate on both DNA and RNA:DNA duplexes. Next, to test the ability of CMG to unwind RNA:DNA hybrids, we constructed oligonucleotide substrates comprising a forked 60 bp duplex in which the template strand is composed of DNA and the non-template strand is composed of either DNA or RNA. Time course analyses demonstrate that CMG unwinds RNA:DNA and DNA:DNA duplexes with similar efficiencies (*Figure 5C*).

Thus, RNA:DNA hybrids on the leading strand at CD R-loops are not automatic blocks to fork progression. In the presence of a 5' flap, the RNA:DNA hybrid can be unwound by CMG, which may account for the generation of full-length replication products observed here in the presence of CD R-loops, and which could also explain the reduction in R-loop levels during CD transcription-replication collisions in human cells (*Hamperl et al., 2017*). In contrast, in the absence of a 5' flap, CMG can translocate on RNA:DNA hybrids without displacing the RNA. Such events may occur at R-loops harboring a flush 5' RNA end or at nicks in the RNA, analogous to replisome encounters with nicks in the lagging strand DNA template (*Vrtis et al., 2021*). Since the RNA:DNA hybrid will form a block to Pol ε, which lacks strand-displacement activity, CMG translocation across RNA:DNA hybrids would induce uncoupling of leading strand synthesis from fork progression, thus providing a molecular basis for the RNase H1-sensitive uncoupling observed at CD R-loops (e.g. native gel, *Figure 4C*, lanes 1 + 2).

## G4s on the template strand can block DNA unwinding by CMG

Next, we tested the impact of G4s on the DNA unwinding activity of CMG. For this, we incorporated a G4 sequence of predicted high stability (G4-wt: 5'-[GGGT]$_3$GGG-3') or a mutant derivative (G4-mut: 5'-GGGTCCCTGGGTGGG-3') in the template or non-template strand of 60 bp forked duplex oligonucleotide substrates. Strikingly, the wildtype G4 sequence, but not its mutant derivative, strongly attenuates DNA unwinding by CMG when placed in the template strand (*Figure 5D*). In contrast, substrate unwinding efficiencies were equivalent in the presence of either wildtype or mutant G4 sequences on the non-template strand (*Figure 5E*). Thus, G4s on the leading strand template have the potential to block DNA unwinding by CMG, explaining the fork stalling at the *Airn* sequence in the presence of HO R-loops or PDS. Consistent with the steric exclusion model for DNA unwinding by CMG (*Kose et al., 2019*), G4s on the lagging strand do not pose an obstacle to fork progression, enabling RNase H1 to promote efficient fork progression at CD R-loops.

## G4s at CD R-loops can induce lagging strand gaps that can be resolved by Pif1

Previous primer extension studies have demonstrated that G4s can impede DNA synthesis by the lagging strand polymerase, Pol δ (*Edwards et al., 2014*; *Sparks et al., 2019b*). If G4s also inhibit progression of the lagging strand polymerase in the context of replisomes is not known. Since the experiments so far were performed in the absence of Fen1 and Cdc9 to allow examination of leading strands, lagging strand blocks were not detectable in the experiments above. Therefore, to investigate the impact of R-loops on lagging strand synthesis, we performed DNA replication reactions in the presence of Fen1 and Cdc9 (*Devbhandari et al., 2017*; *Devbhandari and Remus, 2020*).

In the CD orientation and in the absence of R-loops, replication products obtained in the presence of Fen1/Cdc9 correspond primarily to full-length replication products, indicating that the G-rich strand of the *Airn* sequence does not pose an intrinsic obstacle to lagging strand synthesis (*Figure 6A*, lanes 1–4). In the presence of CD R-loops, leading strand stall and restart products are

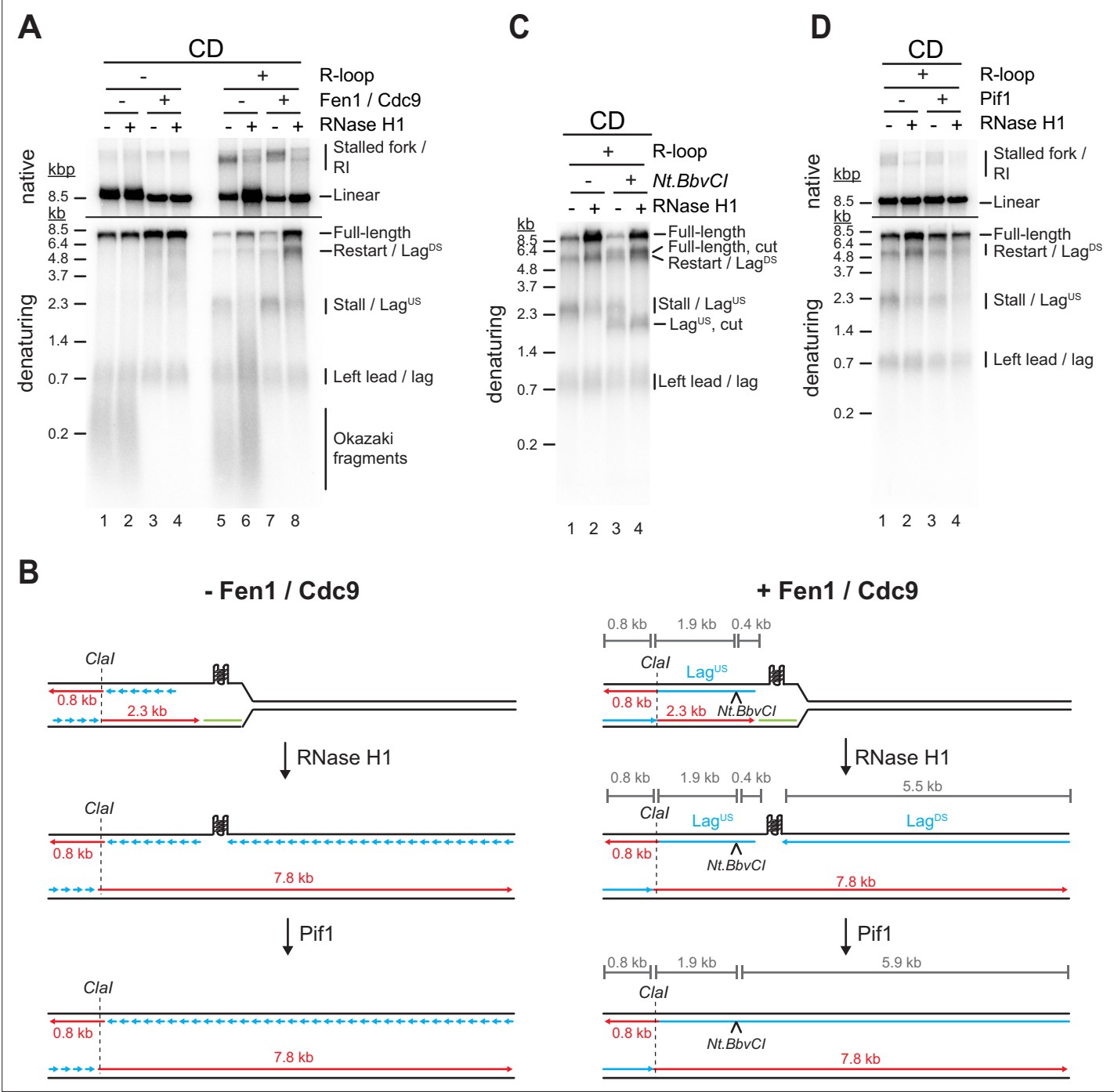

**Figure 6.** G-quadruplexes (G4s) at co-directional (CD) R-loops can induce lagging strand gaps that can be resolved by Pif1. (**A**) Replication products obtained on CD template in the absence or presence of Fen1/Cdc9 and RNase H1. (**B**) Schematic illustrating replication products observed in (**A**). (**C**) Replication products obtained on CD template in the presence of Fen1/Cdc9. RNase H1 was included and products were digested with *Nt.BbvCI* as indicated. (**D**) Replication products obtained on CD template in the presence of Fen1/Cdc9. Pif1 and RNase H1 were included as indicated.

The online version of this article includes the following figure supplement(s) for figure 6:

**Source data 1.** G-quadruplexes (G4s) at co-directional (CD) R-loops can induce lagging strand gaps that can be resolved by Pif1.

**Source data 2.** G-quadruplexes (G4s) at co-directional (CD) R-loops can induce lagging strand gaps that can be resolved by Pif1.

detected as before in the absence of Fen1/Cdc9 (*Figure 6A*, lanes 5 + 7). However, while RNase H1 is sufficient to prevent fork stalling at CD R-loops (*Figure 6A*, lanes 5 + 6; *Figure 4*), in the presence of Fen1/Cdc9 and RNase H1, new replication products are observed at CD R-loops that are identical in size to leading strand stall and restart products (2.3 and 5.5 kb, respectively), but exhibit reversed band intensities (*Figure 6A*, lane 8). These products correspond to lagging strand products generated upstream (Lag$^{US}$) or downstream (Lag$^{DS}$) of the R-loop, respectively (*Figure 6B*). Two lines of evidence are consistent with this interpretation. First, formation of these products is dependent on Okazaki fragment ligation (*Figure 6A*, lanes 6 + 8). Second, Lag$^{US}$, but not the stalled leading strand product of same length, is sensitive to cleavage by the lagging strand specific nicking enzyme *Nt. BbvCI* (*Figure 6B+C*): In the absence of RNase H1, *Nt.BbvCI* digestion separates the 2.3 kb replication products into cleavage-resistant stalled leading strands (2.3 kb) and truncated Lag$^{US}$ (1.9 kb; *Figure 6C*, lanes 1 + 3). In contrast, in the presence of RNase H1, that is, in the absence of fork stalling, the entire population of 2.3 kb products is sensitive to *Nt.BbvCI* cleavage and thus corresponds to Lag$^{US}$ (*Figure 6C*, lanes 2 + 4).

This data is consistent with the notion that R-loop formation induces G4s on the displaced non-template strand, blocking the progression of the lagging strand polymerase during CD replisome collisions, thereby causing a gap in the nascent lagging strand. We, therefore, tested the ability of Pif1 to overcome the lagging strand block at CD R-loops. Indeed, addition of Pif1 greatly reduced the formation of lagging strand gaps at CD R-loops in the presence of RNase H1 (*Figure 6D*, lanes 2 + 4). Thus, Pif1 promotes lagging strand synthesis at G4s in vitro, which is supported by observations in vivo (*Dahan et al., 2018*).

## Both G4s and RNA:DNA hybrids cause lagging strand gaps at HO R-loops

While the G-rich displaced strand forms the lagging strand template at CD R-loops, RNA:DNA hybrids form on the lagging strand template at HO R-loops, presenting a potentially distinct challenge to lagging strand synthesis. Intriguingly, even in the absence of R-loops, a prominent lagging strand gap is observed at the *Airn* sequence in the HO orientation (*Figure 7A*, lanes 1–4; *Figure 7E* i). Restriction analysis confirms the position of Lag$^{US}$ and Lag$^{DS}$ (*Figure 7—figure supplement 1A-C*). The lagging strand gap is caused by G4$^{810-828}$, as deletion of this G4 sequence is sufficient to eliminate the gap (*Figure 7B*, lanes 1–4).

In the presence of R-loops, the length of Lag$^{DS}$ is slightly reduced by treatment with RNase H1 (*Figure 7A*, lanes 7 + 8), indicating that Lag$^{DS}$ stalls at RNA:DNA hybrids during HO R-loop-replisome collisions, but at G4$^{810-828}$ after resolution of the RNA:DNA hybrid by RNase H1 (*Figure 7E* ii). Consistent with RNA:DNA hybrids impeding lagging strand synthesis, HO R-loops cause an RNase H1-sensitive lagging strand gap also on templates in which G4$^{810-828}$ has been deleted (*Figure 7B*, lanes 5–8). As shown above (*Figure 4*), leading strand stalling persists at HO R-loops even in the presence of RNase H1 due to G4s on the G-rich leading strand template, giving rise to stalled leading strand products that are similar in size to Lag$^{US}$ (*Figure 7B*, lane 8; *Figure 7E* iii).

Consistent with the observations above, we find that Pif1 promotes lagging strand synthesis at G4$^{810-828}$ in the absence of R-loops (*Figure 7C*). This Pif1 function is specifically dependent on the Pif1 helicase activity, as mutation of the Pif1 Walker B motif, which has been shown to disrupt the helicase but not DNA binding activity of the human Pif1 orthologue (*Dehghani-Tafti et al., 2019*), abrogates the ability of Pif1 to promote lagging strand synthesis at G4$^{810-828}$ (*Figure 7C*). In contrast, Pif1 alone was unable to promote lagging strand synthesis at RNA:DNA hybrids (*Figure 7D*, lanes 1 + 3) but promoted completion of lagging strand synthesis in conjunction with RNase H1 (*Figure 7D*, lanes 2 + 4). We conclude that both RNA:DNA hybrids and G4s can inhibit lagging strand synthesis at R-loops, and this inhibition is reversed by RNase H1 and Pif1, respectively.

## R-loop transcripts can prime leading strand restart after CD R-loop-replisome collisions

Studies in *Escherichia coli* have demonstrated that transcripts can prime leading strand synthesis after CD collisions of replisomes with RNAP (*Pomerantz and O'Donnell, 2008*). Whether a similar mechanism exists in eukaryotes is unknown. This question is particularly intriguing as CMG tracks along the leading strand template, while the bacterial replicative DNA helicase, DnaB, tracks along the lagging

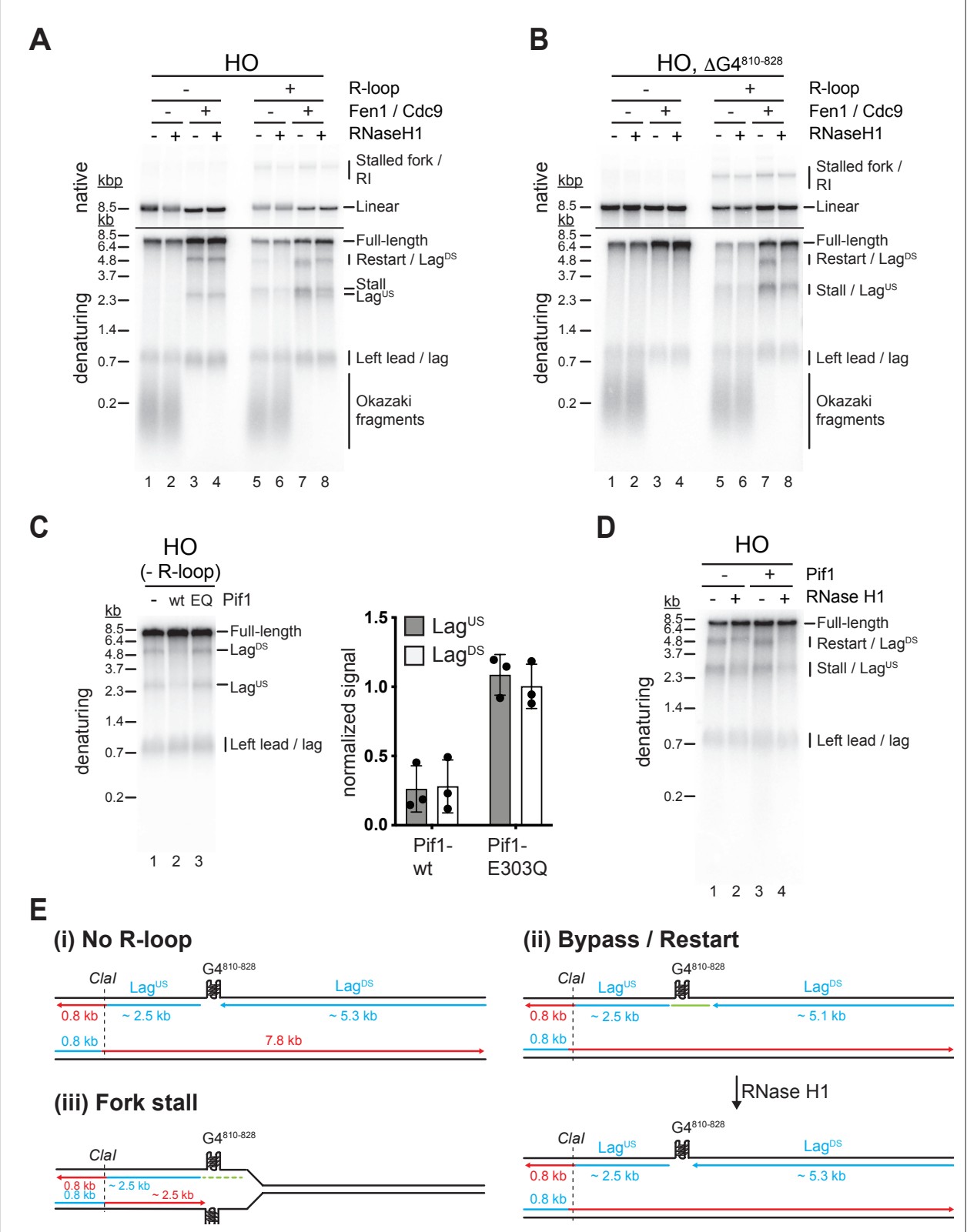

**Figure 7.** Both G-quadruplexes (G4s) and RNA:DNA hybrids cause lagging strand gaps at head-on (HO) R-loops. (**A**) Replication products obtained on HO template in the absence or presence of Fen1/Cdc9 and RNase H1. (**B**) Replication products obtained on HO template harboring ΔG4810-828 deletion in the absence or presence of Fen1/Cdc9 and RNase H1. (**C**) HO template lacking R-loop was replicated with Fen1/Cdc9 in the absence or presence of Pif1 or Pif1-E303Q. Signal intensities of LagUS and LagDS were quantified and normalized to reactions without Pif1. (**D**) HO template replicated with Fen1/

*Figure 7 continued on next page*

*Figure 7 continued*

Cdc9 in the absence or presence of Pif1. (**E**) Schematic illustrating replication products observed in A-D.

The online version of this article includes the following figure supplement(s) for figure 7:

**Source data 1.** Both G-quadruplexes (G4s) and RNA:DNA hybrids cause lagging strand gaps at head-on (HO) R-loops.

**Figure supplement 1.** Both G-quadruplexes (G4s) and RNA:DNA hybrids cause lagging strand gaps at head-on (HO) R-loops.

**Figure supplement 1—source data 1.** Both G-quadruplexes (G4s) and RNA:DNA hybrids cause lagging strand gaps at head-on (HO) R-loops.

strand template. Our purification protocol for R-loop templates involves digestion of free RNA with RNase A. However, RNase A cleavage leaves a 3' phosphate (3'-P), which prevents extension by DNA polymerase. We, therefore, converted RNA 3'-P ends into 3'-hydroxyl (3'-OH) ends using T4 poly-nucleotide kinase (T4 PNK). Strikingly, T4 PNK treatment significantly increased the levels of leading strand restart products at CD R-loops (*Figure 8A*, lanes 1 + 3). Unlike leading strand restart products obtained at CD R-loops in the absence of T4 PNK treatment, these novel restart products are sensitive to RNase H1 treatment (*Figure 8A*). Moreover, formation of these products is dependent on DNA synthesis at replisomes (*Figure 8B*) and is not observed at HO R-loops, in which the RNA:DNA hybrid is on the lagging strand (*Figure 8C*). Together, this data indicates the repriming of leading strand synthesis at the RNA 3'-OH of RNA:DNA hybrids (*Figure 8D*).

We find that under the conditions used here, Pol α, but not Pol δ or Pol ε, is capable of efficiently extending R-loop-associated RNA with DNA (*Figure 8—figure supplement 1A*). Extension by Pol α is limited to ~200–700 nt of DNA and is further increased to up to ~2.5 kb in the presence of all three polymerases, well below the length of complete restart products observed in the context of replisomes. This suggests that leading strand restart at R-loop RNA during CD R-loop-replisome collisions is initiated by Pol α and extended by Pol δ and/or Pol ε. Although Pol δ is not required for leading strand restart at R-loop RNA (*Figure 8—figure supplement 1B*), we do not rule out its involvement, as discussed below.

How may replisomes utilize R-loop transcripts to prime leading strand restart? It is possible that CMG slides over RNA:DNA duplexes at CD R-loops harboring nicks in the RNA, analogous to repli-some encounters with nicks in the lagging strand template (*Vrtis et al., 2021*), which is supported by our demonstration that CMG can translocate on RNA:DNA duplexes (*Figure 5*). To test this model, we treated CD R-loops with sub-saturating concentrations of RNase H1 to introduce nicks in the RNA of CD R-loops. While the formation of full-length leading strand products increases with the concentration of RNase H1 due to the resolution of the RNA:DNA hybrids, leading strand restart correlates inversely with the concentration of RNase H1 (*Figure 8E*). Concomitantly, RNase H1 treat-ment reduces the formation of uncoupled products in the native gel analysis. Similarly, we find that a sub-saturating concentration of purified RNase H2 promotes leading strand restart specifically at CD R-loops (*Figure 8—figure supplement 1C*). We note that RNase H generates a 3'-OH at the cleavage site (*Hyjek et al., 2019*), eliminating the requirement for T4 PNK to convert RNA 3'-P ends into 3'-OH ends for restart. We conclude that R-loop-associated RNAs can prime leading strand restart when CMG can translocate over the RNA:DNA hybrid.

## Discussion

R-loops perform important physiological roles in the cell but can also be harmful if generated in an unscheduled manner. Accordingly, R-loops are considered 'good' or 'bad'. What distinguishes 'good' R-loops from 'bad' ones is unclear. Moreover, although R-loops have been widely recognized as a major determinant of genome instability caused by TRC, the mechanism by which R-loops impede DNA replication has remained obscure. By reconstituting eukaryotic R-loop-replisome collisions in vitro, we demonstrate here that the specific structural configuration of R-loops determines the outcome of R-loop-replisome collisions, providing a mechanistic explanation for 'good' and 'bad' R-loops.

We uncover several ways in which replisomes can continue progression at R-loops. 5' RNA flaps promote the unwinding of RNA:DNA hybrids by CMG on the leading strand. This mechanism is consistent with the observation that CD transcription-replication collisions reduce R-loop levels in cells (*Hamperl et al., 2017*). In the HO orientation, forks may also progress continuously, provided

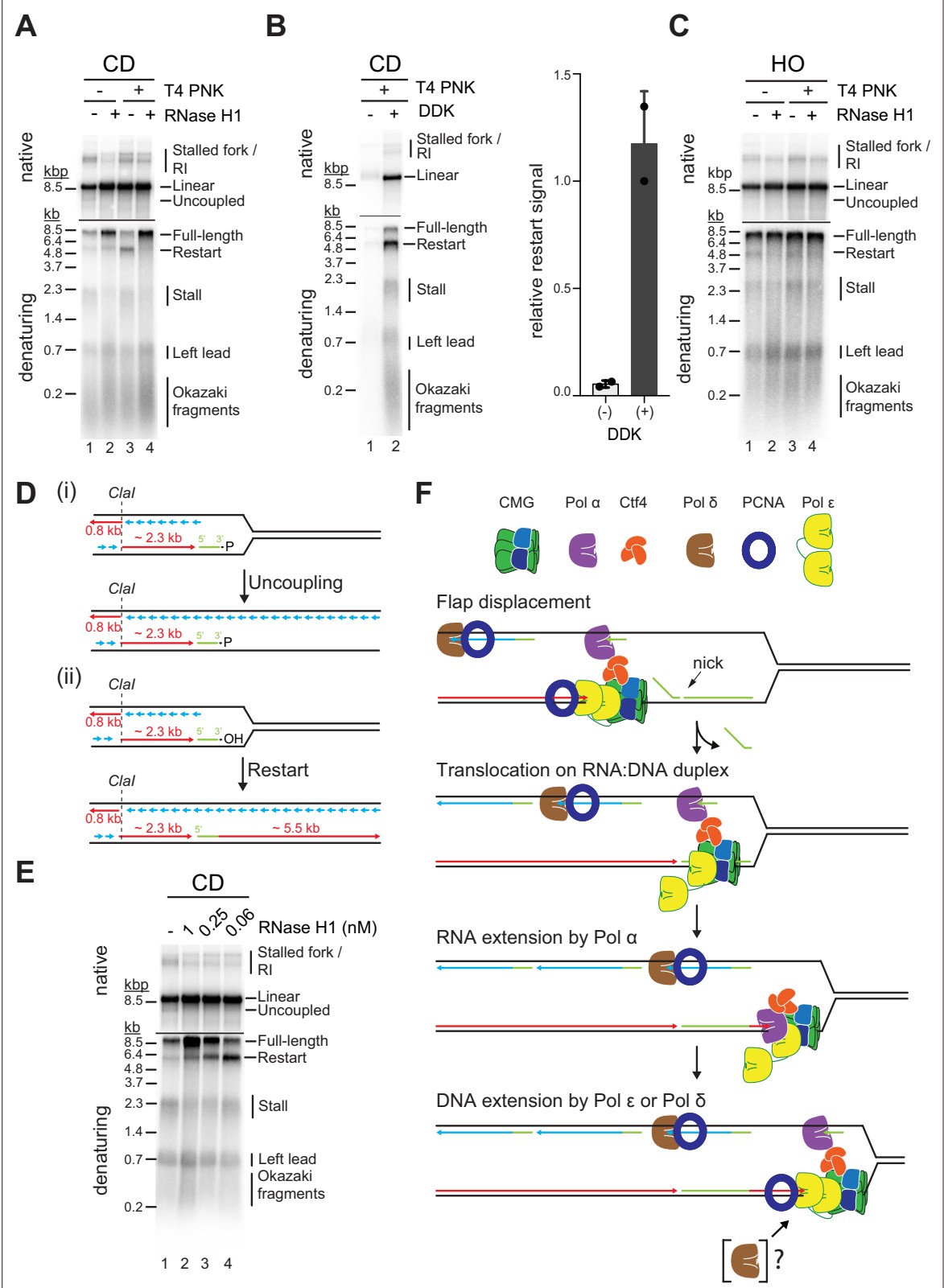

**Figure 8.** R-loop transcripts can prime leading strand restart after co-directional (CD) R-loop-replisome collisions. (**A**) Replication products obtained on mock- or T4 polynucleotide kinase (PNK)-treated CD R-loop-containing templates in absence or presence RNase H1. (**B**) Products obtained on CD T4 PNK-treated R-loop templates in absence or presence of DDK. Relative signal intensity for restart product is quantified on the right. (**C**) Replication products obtained on mock- or T4 PNK-treated CD R-loop-containing templates in absence or presence RNase H1. (**D**) Schematic illustrating replication

*Figure 8 continued*

products observed in A. (**E**) RNase H1 titration into reactions with CD R-loop template. (**F**) Model for leading strand restart at R-loop transcript after replisome encounter with CD R-loop harboring 5′ RNA flap and RNA nick.

The online version of this article includes the following figure supplement(s) for figure 8:

**Source data 1.** R-loop transcripts can prime leading strand restart after co-directional (CD) R-loop-replisome collisions.

**Figure supplement 1.** R-loop transcripts can prime leading strand restart after co-directional (CD) R-loop-replisome collisions.

**Figure supplement 1—source data 1.** R-loop transcripts can prime leading strand restart after co-directional (CD) R-loop-replisome collisions.

**Figure supplement 1—source data 2.** R-loop transcripts can prime leading strand restart after co-directional (CD) R-loop-replisome collisions.

the displaced R-loop strand, which forms the leading strand template for the replisome, is devoid of inhibitory secondary DNA structure. We demonstrate here that G4s can inhibit DNA unwinding by CMG, but it is conceivable that other DNA secondary structures forming on the displaced non-template strand similarly impede fork progression (*Neil et al., 2018*). The ability of replisomes to pass through G4s is likely modulated by G4 stability, which has been shown to influence the impact of G4s on genome stability (*Piazza et al., 2015*).

In addition, we demonstrate that forks can progress through R-loops harboring RNA:DNA hybrids or G4s with polymerase-stalling potential on the leading strand. This results in the uncoupling of leading strand synthesis from replisome progression. Persistent uncoupling would be expected to induce the S phase checkpoint in cells, which may facilitate the clearance of polymerase blocks before S phase completion (*Pardo et al., 2017*; *Saldivar et al., 2017*). Importantly, we also identify distinct mechanisms by which leading strand synthesis can be restarted at G4s and RNA:DNA hybrids.

In the case of G4s, leading strand synthesis is reprimed downstream of the block, causing a gap in the nascent leading strand. Transient uncoupling of leading strand synthesis from fork progression at G4s has been suggested to cause epigenetic instability in vertebrate cells, where repriming is promoted by PrimPol (*Schiavone et al., 2016*). Notably, the leading strand restart observed at G4s here appears to be significantly more efficient than that reported previously at DNA damage sites (*Taylor and Yeeles, 2018*). We speculate that transient stalling of CMG, which may be less pronounced at DNA damage that does not pose a physical obstacle to the CMG, may promote leading strand repriming at G4s.

Our data demonstrates that leading strand restart at RNA:DNA hybrids occurs by a mechanism in which Pol α extends the RNA 3′ end with DNA before handover to the leading strand polymerase at the replisome (*Figure 8F*). To a degree this restart mechanism is analogous to that at DNA damage sites, where restart is initiated by TLS polymerases or Pol δ (*Guilliam and Yeeles, 2021*; *Guilliam and Yeeles, 2020*). The involvement of Pol α can be rationalized by its inherent preference for RNA/DNA junctions (*Perera et al., 2013*). Whether the handover of DNA synthesis from Pol α to Pol ε involves Pol δ at RNA:DNA hybrids remains to be determined. The ability of transcripts to reprime leading strand synthesis was previously observed in *E. coli* (*Pomerantz and O'Donnell, 2008*). However, while the replicative DNA helicase in *E. coli* translocates on the lagging strand, eukaryotic replisomes are assembled around CMG on the leading strand, indicating that the respective mechanisms are fundamentally distinct. We demonstrate that CMG can translocate across RNA:DNA duplexes, allowing the replisome to access RNA 3′ ends of RNA:DNA duplexes. This mechanism requires the RNA 5′ end of RNA:DNA hybrids to be annealed to the template strand in order to prevent unwinding by the replisome. The structure of R-loops in vivo is not known and is likely variable. However, we note that R-loop formation may be promoted by invasion of the DNA duplex by the RNA 5′ end in order to limit topological clashes during the winding of the RNA around the template strand (*Belotserkovskii et al., 2018*). Alternatively, flush RNA 5′ ends may be encountered by the replisome at nicks in the RNA, analogous to lagging strand nicks (*Vrtis et al., 2021*). Such nicks may be generated by RNase H1 or RNase H2, whose activities are spatially and temporally limited, respectively (*Lockhart et al., 2019*; *Zimmer and Koshland, 2016*). It is noteworthy that translocation of the CMG on dsDNA during replication termination or after encounter of lagging strand nicks induces ubiquitin-dependent replisome disassembly (*Deegan et al., 2020*; *Vrtis et al., 2021*). It may, therefore, be interesting to test if CMG translocation on the RNA:DNA hybrid at CD R-loops can cause analogous replisome disassembly, which would compete with fork restart and promote fork collapse instead. However, we also note that the lagging strand template, which protects the CMG

from ubiquitylation during normal fork progression, remains accessible at CD R-loops, contrary to the situation during termination or replisome encounter of lagging strand nicks, which may limit any potential replisome disassembly.

We find that both RNA:DNA hybrids and G4s can also induce gaps in nascent lagging strands. This is consistent with lagging strand G4s impeding DNA replication in vivo and with G4s posing a block to Pol δ in primer extension assays (*Dahan et al., 2018*; *Sparks et al., 2019b*). However, the block to lagging strand synthesis by RNA:DNA hybrids is surprising given the proficiency of the lagging strand machinery in removing RNA:DNA hybrids at Okazaki fragment 5' ends (*Stodola and Burgers, 2017*). The reasons for this are currently unknown, but it is possible that long RNA 5' flaps or G4s in the RNA strand impinge on the ability of Pol δ and Fen1 to process RNA:DNA hybrids.

We show that RNA:DNA hybrid- and G4-induced blocks to fork progression, as well as leading and lagging strand synthesis, are mitigated by RNase H1 and Pif1, respectively. This data is consistent with in vivo studies demonstrating that both RNase H1 and Pif1 suppress genome instability at R-loops (*Huertas and Aguilera, 2003*; *Li and Manley, 2005*; *Tran et al., 2017*) and promote replication fork progression at R-loops and G4 sequences, respectively (*Gan et al., 2011*; *Paeschke et al., 2011*; *Prado and Aguilera, 2005*). We note that the coincidence of RNA:DNA hybrids and G4s at R-loops may limit the efficiency of RNase H1 overexpression to resolve R-loops, which is commonly used to assess the contribution of R-loops to TRC.

R-loops in both the CD and HO orientation can induce persistent fork stalling, which requires the RNA:DNA hybrid- and/or G4-resolving activities of RNase H1 and Pif1 for continued fork progression. However, we also find that replisomes have an intrinsic capacity to progress through R-loops by bypassing or unwinding G4s or RNA:DNA hybrids. Despite the potential to induce nascent strand gaps, continued fork progression may be beneficial in order to establish homology-directed repair-competent chromatin on sister chromatids (*Nakamura et al., 2019*; *Saredi et al., 2016*). Nascent strand discontinuities at R-loops may also be processed post-replicatively (*Wu et al., 2020*), which could allow the timely progression of S phase. Alternatively, replisome bypass of R-loops may promote the resolution of R-loops analogous to the repair of DNA-protein cross-links (*Sparks et al., 2019a*).

The molecular characterization of R-loop-replisome collision presented here enhances our understanding of the mechanism(s) by which G4 ligands can affect the growth of cancer cells (*De Magis et al., 2019*). Moving forward, we expect that the system developed here will aid future studies directed at characterizing the molecular functions of the multitude of additional factors implicated in the processing of RNA:DNA hybrids and G4s (*Garcia-Muse and Aguilera, 2019*; *Lerner and Sale, 2019*). While the experiments reported here were designed to specifically assess the impact of the R-loop nucleic acid structure on replisome progression, many additional factors, such as RNAP, transcription factors, and specialized chromatin structures, are associated with R-loops in vivo and further influence the outcome of R-loop-replisome collisions (*Castellano-Pozo et al., 2013*; *Garcia-Muse and Aguilera, 2019*; *Garcia-Pichardo et al., 2017*; *Garcia-Rubio et al., 2018*). Reconstitution of R-loop-replisome collisions on chromatin templates will thus be an interesting challenge for future studies. Moreover, since DNA supercoiling affects R-loop formation

**Table 1.** Yeast strains.

| Strain name | Protein | Genotype |
| --- | --- | --- |
| YSD35 | CMG | MAT a/α *ade2-1 ura3-1 his3-11,15 trp1-1 leu2-3,112 can1-100 pep4::kanMX bar1::hphNAT1* Gal-*Gal4* (HIS3) Gal-*Mcm5/Mcm4* (TRP1) Gal-*Mcm2*/FLAG-*Mcm3* (URA3) Gal-*Mcm7/Mcm6* (LEU2) Gal1-10 *Cdc45* (TRP1) Gal1-10 *Psf2++/Psf3++* (URA3) Gal1-10 *Psf1++/*CBP-*Sld5++* (LEU2) |
| YDR154 | RNase H2 | *MATa ade2-1 ura3-1 his3-11,15 trp1-1 leu2-3,112 can1-100 pep4::kanMX bar::hphNAT1* Gal-*Gal4* (HIS3) GAL-CBP-*RNH201++* (LEU2) GAL-*RNH202++ / RNH203++* (URA3) |

and stability (*Chedin and Benham, 2020*), it will be of interest to investigate the impact of R-loops on replication fork progression in topologically closed DNA templates.

# Materials and methods

## Key resources table

| Reagent type (species) or resource | Designation | Source or reference | Identifiers | Additional information |
|---|---|---|---|---|
| Strain, strain background (*Saccharomyces cerevisiae*) | YDR35 | This study | See *Table 1* for genotype | Overexpression and purification of CMG |
| Strain, strain background (*Saccharomyces cerevisiae*) | YDR154 | This study | See *Table 1* for genotype | Overexpression and purification of RNase H2 |
| Strain, strain background (*Saccharomyces cerevisiae*) | ySD-ORC | PMID:23474987 | | Overexpression and purification of ORC |
| Strain, strain background (*Saccharomyces cerevisiae*) | YJF38 | PMID:23474987 | | Overexpression and purification of Cdt1·Mcm2-7 |
| Strain, strain background (*Saccharomyces cerevisiae*) | YSA35 | PMID:32701054 | | Overexpression and purification of DDK |
| Strain, strain background (*Saccharomyces cerevisiae*) | YSD15 | PMID:27989437 | | Overexpression and purification of Cdc45 |
| Strain, strain background (*Saccharomyces cerevisiae*) | YDR105 | PIMD:27989437 | | Overexpression and purification of Clb5·Cdk1 |
| Strain, strain background (*Saccharomyces cerevisiae*) | YDR109 | PIMD:27989437 | | Overexpression and purification of GINS |
| Strain, strain background (*Saccharomyces cerevisiae*) | YSD116 | PIMD:27989437 | | Overexpression and purification of Pol ε |
| Strain, strain background (*Saccharomyces cerevisiae*) | YDR110 | PIMD:27989437 | | Overexpression and purification of Dpb11 |
| Strain, strain background (*Saccharomyces cerevisiae*) | YSD13 | PIMD:27989437 | | Overexpression and purification of Sld2 |
| Strain, strain background (*Saccharomyces cerevisiae*) | YSD16 | PIMD:27989437 | | Overexpression and purification of Pol α |
| Strain, strain background (*Saccharomyces cerevisiae*) | YIW389 | PIMD:27989437 | | Overexpression and purification of RFC |
| Strain, strain background (*Saccharomyces cerevisiae*) | YDR131 | PIMD:27989437 | | Overexpression and purification of Pol δ |
| Strain, strain background (*Saccharomyces cerevisiae*) | YDR128 | PIMD:27989437 | | Overexpression and purification of Top1 |
| Strain, strain background (*Saccharomyces cerevisiae*) | YDR163 | PIMD:32341532 | | Overexpression and purification of Mrc1 |
| Strain, strain background (*Saccharomyces cerevisiae*) | YDR137 | PIMD:32341532 | | Overexpression and purification of Csm3-Tof1 |
| Antibody | S9.6 (mouse monoclonal) | Sigma Millipore | MABE1095 | (1 µg per 1 µg) of DNA |
| Recombinant DNA reagent | pFC57 | PMID:22387027 | | *AIRN*-containing plasmid |
| Recombinant DNA reagent | p470 | PMID:27989437 | | *ARS305*-containing plasmid |
| Recombinant DNA reagent | p1216 | This paper | | p470 with secondary origin sequences deleted, without *AIRN* |
| Recombinant DNA reagent | P1214 | This paper | | p1216 with *AIRN* in CD orientation |
| Recombinant DNA reagent | p1215 | This paper | | p1216 with *AIRN* in HO orientation |

*Continued on next page*

*Continued*

| Reagent type (species) or resource | Designation | Source or reference | Identifiers | Additional information |
|---|---|---|---|---|
| Recombinant DNA reagent | p1290 | This paper | | p1214 with BbvCI site 2075 bp downstream of *AIRN* |
| Recombinant DNA reagent | p1285 | This paper | | p1215 with BbvCI site 2075 bp downstream of *AIRN* |
| Recombinant DNA reagent | p1291 | This paper | | p1214 with BbvCI site 11 bp upstream of *AIRN* |
| Recombinant DNA reagent | p1292 | This paper | | p1215 with BbvCI site 11 bp upstream of *AIRN* |
| Recombinant DNA reagent | p1293 | This paper | | CD, *AIRN*Δ810–828 |
| Recombinant DNA reagent | p1284 | This paper | | HO, *AIRN*Δ810–828 |
| Recombinant DNA reagent | p1134 | This paper | pET15b-6x-His-RNH1 | Overexpression of His-tagged Rnh1 |
| Recombinant DNA reagent | p1254 | This paper | pET15b-T7 Lysozyme-6x-His | Overexpression of C-terminally His-tagged T7 lysozyme |
| Recombinant DNA reagent | p1102 | This paper | pET15b-PIF1-N | Overexpression of His-tagged Pif1-N |
| Recombinant DNA reagent | p1130 | This paper | pET15b-PIF1-N-E303Q | Overexpression of His-tagged Pif1-N-E303Q |
| Recombinant DNA reagent | pET15b-His-CDC6 | PMID:24566988 | | Overexpression of His-tagged Cdc6 |
| Recombinant DNA reagent | p399 | PIMD:27989437 | pSmt3-SLD3 | Overexpression of His-tagged Sld3 |
| Recombinant DNA reagent | P468 | PIMD:27989437 | pGEX-6P-1-SLD7 | Overexpression of GST-tagged - Sld7 |
| Recombinant DNA reagent | pJM126 | PIMD:27989437 | | Overexpression of RPA |
| Recombinant DNA reagent | pCK6 | PIMD:27989437 | pET15b-His-CTF4 | Overexpression of His-tagged Ctf4 |
| Recombinant DNA reagent | p1012 | PIMD:27989437 | pET28a-PCNA | Overexpression of PCNA |
| Recombinant DNA reagent | p619 | PIMD:27989437 | pET15b-His-MCM10 | Overexpression of His-tagged Mcm10 |
| Recombinant DNA reagent | p1137 | PMID:32341532 | pET28a-FEN1-6xHis | Overexpression of C-terminally His-tagged Fen1 |
| Recombinant DNA reagent | p1019 | PMID:32341532 | pET28a-CDC9 | Overexpression of His-tagged Cdc9 |
| Sequence-based reagent | DNA oligo-nucleotides | IDT | (See *Table 2 Table 3*) | To generate templates for helicase assays |
| Chemical compound, drug | Pyridostatin | Sigma Aldrich | SML2690 | |
| Chemical compound, drug | ATP | Thermo Scientific | R1441 | |
| Chemical compound, drug | Formamide | ThermoFisher Scientific | 17899 | |
| Chemical compound, drug | Benzalkonium chloride | MilliporeSigma | B6285 | |
| Peptide, recombinant protein | RNase H | New England Biolabs | M0297S | |
| Peptide, recombinant protein | T7 RNA polymerase | Promega | P2075 | |
| Peptide, recombinant protein | RNase A | ThermoFisher Scientific | EN0531 | |
| Peptide, recombinant protein | Proteinase K | New England Biolabs | P8107S | |

*Continued on next page*

*Continued*

| Reagent type (species) or resource | Designation | Source or reference | Identifiers | Additional information |
|---|---|---|---|---|
| Peptide, recombinant protein | T4 polynucleotide kinase | New England Biolabs | M0201S | |
| Software, algorithm | GraphPad Prism | GraphPad Prism (https://graphpad.com) | RRID:SCR_015807 | Statistical analysis and data plotting |
| Software, algorithm | ImageJ | ImageJ (https://imagej.nih.gov/ij/) | RRID:SCR_003070 | Image analysis |
| Software, algorithm | QGRS mapper | QGRS mapper (https://bioinformatics.ramapo.edu/QGRS/analyze.php) | | Prediction of G4 sequences |

DNA templates for replication assays p1214 and p1215 were generated from pARS305 (*Gros et al., 2014*). Cryptic origins were eliminated by excising the region flanked by *ClaI* and the region flanked by *BaeI* and *BsaAI*. The *Airn* sequence was amplified from pFC53 (kind gift from Fred Chedin) (*Ginno et al., 2012*). T7 promoter, 3 × T7 terminators and the *Airn* sequence were inserted at the *BsaAI* site by Gibson assembly. p1290 and p1285 were generated from p1214 and p1215, respectively, by inserting a *BbvCI* recognition sequence between *BglII* and *SwaI* sites, 2075 bp downstream of the *Airn* sequence. p1291 was generated from p1214 by inserting a *BbvCI* recognition sequence at the *BstZ17I* site. p1292 was generated from p1215 by inserting a *BbvCI* recognition sequence at the *ApaI* site. To generate p1293 and p1284, the *Airn* sequence at position 810–828 was deleted by PCR mutagenesis and the mutant *Airn* sequence inserted at the *BsaAI* site of p1214 and p1215, respectively.

## R-loop templates

Transcription reactions were carried out for 20 min at 37 °C in a reaction volume of 250 μL containing 20 μg of plasmid template in 1 × transcription buffer (Promega; 40 mM Tris-HCl pH 7.9/6 mM MgCl$_2$ /10 mM spermidine/50 mM NaCl)/25 U T7 RNA Polymerase (Promega)/20 mM DTT/0.05 % Tween 20/0.5 mM each ATP, GTP, CTP, and UTP. Subsequently, 0.3 μM T7 lysozyme were added to the reaction and incubation continued for 10 min at 37 °C. Then, the salt concentration was adjusted to 400 mM NaCl, 10 μg/mL RNase A (ThermoScientific) were added to the reaction, and incubation continued for 30 min at 37 °C. Finally, 4 U of Proteinase K (NEB) were added to the reaction and incubation continued for 30 min at 37 °C. The reaction was concentrated to a volume of 50 μL using Amicon Ultra 0.5 mL centrifugal filter and fractionated through a custom-made 2 mL S1000 column equilibrated in 10 mM Tris-HCl pH 7.5. Template-containing peak fractions were pooled, concentrated, and stored at –20 °C. $^{32}$P-RNA-labeled R-loop templates were generated as above, with the exception that transcription reactions were performed in the presence of 0.25 mM GTP and 60 μCi of α-[$^{32}$P]-GTP (Perkin Elmer).

## Calculation of G/C skew

GC skew at every nucleotide of the 1374 bp *Airn* sequence was calculated using a 100-nucleotide sliding window as described (*Ginno et al., 2012*). Briefly, the number of Gs and Cs was counted in each given window. The G/C skew was calculated as (G-C)/(G + C).

## Templates for helicase assays

Individual oligonucleotides for helicase template preparation (*Table 3*) were gel-isolated using the crush and soak method: Oligonucleotides were electrophoresed on 8 % denaturing polyacrylamide gels at 120 V for 1.5 hr in 1 × TBE buffer. The bands were excised, crushed into a fine paste, and soaked overnight in an Eppendorf ThermoMixer at 37 °C and 1400 rpm in 500 mM ammonium acetate/10 mM magnesium acetate/1 mM EDTA. The extracted oligonucleotides were ethanol-precipitated twice and stored in TE pH 8.0.

Radiolabeling of oligonucleotides was carried out for 1 hr at 37 °C using T4 PNK (NEB), 0.25 μM oligonucleotide, and 0.5 μM γ-[$^{32}$P]-ATP. Reactions were terminated by incubation for 20 min at 80 °C. Following the radiolabeling, oligonucleotide annealing was carried out in a thermocycler in T4 PNK

reaction buffer (NEB)/50 mM potassium acetate by heating oligonucleotide mixtures to 95 °C for 5 min, followed by cooling at a rate of 1 °C/min until the temperature reached 10 °C. Annealed products were fractionated by 10 % native PAGE at 150 V for 45 min in 0.5 × TBE, corresponding bands excised from the gel, and templates eluted by soaking the excised gel slices overnight in buffer containing 10 mM Tris-HCl pH 7.2 (for templates with RNA) or pH 8.0 (for templates with DNA)/50 mM potassium acetate/1 mM EDTA.

## Gel-analysis of R-loop templates

For *Figure 1D*, 0.5 μg of R-loop-containing plasmid was incubated with 0.5 U RNase H (NEB) in 75 mM KCl/50 mM Tris-HCl pH 7.5/3 mM MgCl$_2$ /10 mM DTT in a 20 μL reaction volume and incubated at 37 °C for 30 min. The reaction was quenched with 40 mM EDTA. Half the reaction was added to loading dye (10 mM Tris-HCl pH 7.6/0.15 % orange G/60 % glycerol/60 mM EDTA) and analyzed by 1 % agarose gel-electrophoresis in TAE. The gel was run at 40 V for 5 hr and stained with ethidium bromide.

For *Figure 1E*, 0.5 μg of $^{32}$P-labeled R-loop-containing plasmid was either mock-treated or incubated with 0.5 U RNase H (NEB) in 75 mM KCl/50 mM Tris-HCl pH 7.5/3 mM MgCl$_2$ /10 mM DTT in a 20 μL reaction volume for 30 min at 37 °C. The reaction was quenched with 40 mM EDTA. Half the reaction was added to denaturing loading buffer (50 % formamide/6 % formaldehyde/10 % glycerol/0.01 % bromophenol blue) and incubated for 2 min at 90 °C, cooled on ice, and fractionated on a 1 % agarose gel containing 0.7 % formaldehyde. The gel was washed in 1 × SSC, dried on Whatman paper, and analyzed by phosphor imaging.

## Electron microscopy

Plasmid templates for all EM analyses were linearized with *MscI* (NEB).

### T4 gene 32 protein (gp32) staining

To stain the displaced single strand (ss) DNA of the R-loop with gp32 (gift of Nancy Nossal, NIH, Bethesda, MD), linearized template plasmid was mixed with gp32 at a ratio of 1 μg of gp32 per μg of DNA in a binding buffer of 10 mM Hepes pH 7.5/1 mM EDTA/50 mM NaCl on ice for 30 min. The sample was then fixed with 0.6 % glutaraldehyde for 5 min on ice followed by surface spreading the DNA on a droplet of 0.25 mM ammonium acetate, with 7 μg/mL of cytochrome C protein and allowed to develop for 3 min followed by picking up the surface film with a Parlodion coated 400 mesh copper grids. The grids were then rotary shadow cast with 80 % platinum–20 % palladium (Electron Microscopy Sciences) followed by an overlayer of carbon.

### S9.6 antibody staining

Linearized template plasmid DNA was mixed with the S9.6 antibody at a ratio of 1 μg of antibody per μg of DNA in the binding buffer above, fixed, and prepared for EM as described for gp32 staining.

### Formamide spreading

DNA spreads were prepared according to *Lopes, 2009*. Typically, 5 μL of DNA corresponding to 5–20 ng were used for each spread. The DNA was mixed with 5 μL of formamide and 0.4 μL of 0.02 % benzalkonium chloride in 10 mM Tris, 1 mM EDTA pH 7.5. After mixing, the drop was immediately spread on a water surface in a 15 cm dish containing 50 mL of distilled water, using a freshly cleaved mica sheet (Ted Pella Inc) as a ramp. The monomolecular layer was gently touched with a carbon-covered 400-mesh copper grid, activated just before use by floating it on a drop of ethidium bromide solution (33.3 μg/mL in H$_2$O), for 30–45 min at room temperature. Grids with adsorbed DNA molecules were immediately stained with a solution of 0.2 μg/μL uranyl acetate in ethanol and transferred to a Denton evaporator where the sample was rotary shadow cast with platinum at a vacuum of 2 × 10$^{-6}$ Torr and at an angle of 3° between the sample and the platinum wire.

### Distance measurements

ImageJ was used to measure the distance from the short end of the DNA to the start of the loop. At least 100 images each for CD and HO templates were used for distance measurements. Distances were plotted and frequency distribution analyses were performed using GraphPad Prism.

**Table 2.** List of oligonucleotides used to prepare templates for helicase assays.

| Name | Sequence 5' to 3' |
| --- | --- |
| A | GGCTCGTTTTACAACGTCGTGCTGAGGTGATATCTGCTGAGGCAATGGGAATTCGCCAACCTTTTTTTTTTTTTTTTTTTTTTTTTTTTTT |
| B | GGCAGGCAGGCAGGCAGGCAGGCAGGTTGGCGAATTCCCATTGCCTCAGCAGATATCACCTCAGCACGACGTTGTAAAACGAG |
| C | GGCAGGCAGGCAGGCAGGCAGGCAGGCAGGCAGGTTGGCGAAUUCCCAUUGCCUCAGCAGAUAUCACCUCAGCAGAUAUCACCUCAGCAGUUGUAAAACGAG |
| D | GGCAGGCAGGCAGGCAGGCAGGCAGGCAGGCAGATTAAGTTGGGTAACGCCAGGGTTTCCCAGTCACGAC |
| E | GTCGTGACTGGGAAAACCCTGGCGTTACCCAACTTAATCGCCTTGCAGCACATCCCCCTTTGCGCCAGCTGGCGTAAATAGTTTTTTTTTTTTTTTTTTTTTTTTTTTTTTTTTTTTTTTTT |
| F | CTATTACGCCAGCTGGCGAAAGGGGGATGTGCTGCAAGGC |
| G | CUAUUACGCCAGCUGGCGAAAGGGGGAUGUGCUGCAAGGC |
| H | GGCAGGCAGGCAGGCAGGCAGGCAGGCAGGTTGGCGAATTCCCTTTTTTTTTTTTTTTTCCTCAGCACGACGTTGTAAAACGAG |
| I | GGCTCGTTTTACAACGTCGTGCTGAGGTTGGGTGGGTTGGGAATTCGCCAACCTTTTTTTTTTTTTTTTTTTTT |
| J | GGCTCGTTTTACAACGTCGTGCTGAGGTTGGGTGGGTCCCTGGGTGGGTTGGGAATTCGCCAACCTTTTTTTTTTTTTTTTTTTTT |
| K | GGCTCGTTTTACAACGTCGTGCTGAGGTTTTTTTTTTTGGGAATTCGCCAACCTTTTTTTTTTTTTTTTTTTTTT |
| L | GGCAGGCAGGCAGGCAGGCAGGCAGGTTGGCGAATTCCCTTGGGTGGGTGGGTTCCTCAGCACGACGTTGTAAAACGAG |
| M | GGCAGGCAGGCAGGCAGGCAGGCAGGCAGGTTGGCGAATTCCCTTGGGTGGGTGGGTTCCTCAGCACGACGTTGTAAAACGAG |

**Table 3.** List of templates used in helicase assays.

| Template | Oligonucleotides | Figure | [32]P-labeled oligonucleotide |
|---|---|---|---|
| Forked DNA duplex | A + B | 4 C, left | A |
| Forked RNA-DNA duplex | A + C | 4 C, right | A |
| DNA-DNA 1 | D + E + F | 4B(i) | D |
| DNA-DNA 2 | D + E + F | 4B(iii) | F |
| RNA-DNA 1 | D + E + G | 4B(ii) | D |
| RNA-DNA 2 | D + E + G | 4B(iv) | G |
| G-quad wt, lead | H + I | 4D, left | H |
| G-quad mut, lead | H + J | 4D, right | H |
| G-quad wt, lag | K + L | 4E, left | K |
| G-quad mut, lag | K + M | 4E, right | K |

## Atomic force microscopy

A 50 µL reaction containing 10 nM linearized R-loop-containing plasmid/22 nM RPA/5 % glycerol/0.3 mM ATP/10 mM Mg(OAc)$_2$/100 mM KOAc/25 mM Hepes-KOH pH 7.5 was incubated on ice for 3 min; 40 µL of this reaction were deposited onto freshly cleaved mica for 2 min. The sample was rinsed with 10 mL ultrapure deionized water and the surface was dried using a stream of nitrogen. AFM images were captured using an Asylum Research MFP-3D-BIO (Oxford Instruments) microscope in tapping mode at room temperature. An Olympus AC240TS-R3 AFM probe with resonance frequencies of approximately 70 kHz and spring constant of approximately 1.7 N/m was used for imaging. Images were collected at a speed of 0.5–1 Hz with an image size of 2–3 µm at a 2048 × 2048 pixel resolution.

## Proteins

Unless specified below, proteins were purified as described previously (***Devbhandari et al., 2017***; ***Devbhandari and Remus, 2020***).

## T7 lysozyme

T7 lysozyme was expressed as C-terminally His-tagged fusion protein in BL21 DE3 RIL cells; 1 L of cells was grown to log phase, induced with 1 mM IPTG at 37°C for 3 hr. Cells were harvested by centrifugation for 30 min at 4000 rpm and resuspended in buffer L (50 mM Tris-HCl pH 7.6/10 % glycerol/0.02% NP-40)/300 mM NaCl/protease inhibitor cocktail/1 mM DTT. Cells were lysed by incubation with 0.1 mg/mL lysozyme for 30 min on ice followed by sonication. Cleared lysate was obtained by centrifugation at 15,000 rpm in an SS34 rotor for 30 min at 4 °C. The clarified extract was supplemented with 10 mM imidazole and rotated for 2 hr at 4 °C with 1 mL of Ni-NTA beads (Qiagen). Bound protein was eluted with 10 CV of buffer L/200 mM imidazole. Peak fractions were pooled, diluted with buffer L to lower the salt concentration to 10 mM NaCl, and fractionated over a 1 mL MonoS column using a 20 CV gradient of 10–125 mM NaCl in buffer L. Peak fractions were pooled, concentrated using Amicon Ultra 4 mL centrifal filters (Millipore) and fractionated over 24 mL S200 gel filtration column equilibrated in buffer L/125 mM NaCl. Peak fractions were pooled, aliquoted, flash-frozen, and stored at –80 °C.

## RNase H1

RNase H1 was expressed as an N-terminal 6x-His tag fusion protein in BL21 DE3 RIL cells; 1 L of cells was grown to log phase and induced with 1 mM IPTG at 16 °C overnight. Cells were harvested by centrifugation and resuspended in buffer A (25 mM potassium phosphate pH 7.5/10 % glycerol/0.02% NP-40)/500 mM KCl/1 mM DTT/protease inhibitors. Cells were lysed by addition of 0.1 mg/mL lysozyme, incubation for 30 min on ice, followed by sonication. Insoluble material was removed by centrifugation at 15,000 rpm in an SS35 rotor for 30 min at 4 °C. Clarified extract was supplemented with 10 mM

imidazole, passed over 1 mL Ni-NTA resin (Qiagen), and bound protein eluted with 10 CV of buffer A/200 mM imidazole. Peak fractions were pooled, dialyzed against buffer B (25 mM Hepes-KOH pH 7.5/10 % glycerol/1 mM EDTA/1 mM EGTA/0.02% NP-40)/0.1 M KCl/1 mM DTT, and fractionated over a 1 mL MonoS column using a salt gradient of 0.1–1 M KCl over 20 CV. Peak fractions were pooled, concentrated in an Amicon Ultra 4 mL centrifugal filter, and fractionated over a 24 mL S200 gel filtration column equilibrated in buffer B/0.3 M KOAc/1 mM DTT. Peak fractions were pooled, aliquoted, flash-frozen, and stored at –80 °C.

## RNase H2

Strain YDR154 was grown at 30 °C in YP-GL (YP + 2 % glycerol/2 % lactic acid) to a density of $5 \times 10^7$ cells/mL, and protein expression was induced by addition of 2 % galactose for 4 hr. Cells were harvested by centrifugation and washed once with cold 25 mM Hepes-KOH pH 7.6/1 M sorbitol, and once with with buffer C (25 mM Hepes-KOH pH 7.6/0.02 % NP-40/10 % glycerol/1 mM DTT)/100 mM KCl. The cell pellet was resuspended in 0.5 volumes of buffer C/100 mM KCl/protease inhibitor, and frozen dropwise in liquid nitrogen. The frozen cell popcorn was crushed in a freezer mill (SPEX Certi-Prep 6850 Freezer/Mill) for 10 cycles of 2 min at a rate of 15 impacts per second. The crushed cell powder was thawed on ice and mixed with 1 volume of buffer C/100 mM KCl/protease inhibitor. The KCl concentration was adjusted to 0.4 M and the cell lysate rotated for 30 min at 4°C. Insoluble material was precipitated by ultracentrifugation for 1 hr at 40,000 rpm in a T647.5 rotor (Thermo Scientific). The clarified cell lysate was supplemented with 2 mM $CaCl_2$ and rotated for 2–3 hr at 4°C with 1 mL of calmodulin affinity resin. The resin was collected in a disposable column, washed with 10 CV of buffer C/2 mM $CaCl_2$ /500 mM KCl, and bound protein eluted in 6 CV of buffer C/500 mM KCl/1 mM EDTA/2 mM EGTA. Peak fractions were pooled, dialyzed against buffer C/100 mM KCl/1 mM EDTA/1 mM EGTA, and fractionated on a 1 mL MonoS column using a gradient of 0.1–1 M KCl in buffer C/1 mM EDTA/1 mM EGTA. Peak fractions were pooled and gel-filtered on a 24 mL S200 column equilibrated in buffer C/0.3 M KOAc/1 mM EDTA/1 mM EGTA/1 mM DTT. Peak fractions were pooled and stored in aliquots at –80°C.

## Pif1

The nuclear form (amino acids 40–859) of Pif1 was expressed as an N-terminally 6x-His tagged fusion protein in BL21 DE3 RIL cells; 2 L of cells were grown to log phase and induced with 0.1 mM IPTG for 18 hr at 16 °C. Cells were harvested by centrifugation, resuspended in 50 mL buffer A/100 mM KCl/1 mM DTT/protease inhibitors, and cells lysed by addition of 0.1 mg/mL lysozyme, incubation for 30 min on ice, and subsequent sonication. Lysate was clarified by centrifugation in an SS34 rotor at 15,000 rpm for 30 min at 4 °C. Clarified extract was passed over a 5 mL Q sepharose column, the flow-through applied to a 5 mL SP-Sepharose column, and bound protein eluted with a salt gradient of 150–660 mM KCl in buffer A over 10 CV. Subsequent purification steps were as described before (*Wilson et al., 2013*). Pif1-N-E303Q was purified as described above.

## CMG

Strain YSD35 was grown at 30 °C in YP-GL (YP +2 % glycerol/2 % lactic acid) to a density of $5 \times 10^7$ cells/mL, and protein expression was induced by addition of 2 % galactose for 6 hr. Cells were harvested by centrifugation and washed once with 25 mM HEPES-KOH pH-7.6/1 M sorbitol and once with buffer D (40 mM HEPES-NaOH pH-7.6/150 mM sodium acetate/10 % glycerol/0.005 % Tween 20). The cell pellet was resuspended in 0.5 volumes of buffer D/1 mM DTT/protease inhibitor, and frozen dropwise in liquid nitrogen. The frozen cell popcorn was crushed in a freezer mill (SPEX CertiPrep 6850 Freezer/Mill) for 10 cycles of 2 min at a rate of 15 impacts per second. The crushed cell powder was thawed on ice and mixed with 0.5 volumes of buffer D/1 mM DTT. The cell lysate was centrifuged for 1 hr at 40,000 rpm in T647.5 rotor (Thermo Scientific) to sediment insoluble material. The clarified extract was passed over 6 mL of FLAG M2-affinity beads (Sigma) using the Biorad Econo pumping system. The beads were washed with 10 CV of buffer D/1 mM DTT, followed by washes with 20 mL of buffer D/1 mM DTT/5 mM magnesium acetate/0.5 mM ATP and 10 CV of buffer D/1 mM DTT. Bound protein was eluted with 1 CV of buffer D/1 mM DTT/2 mM $CaCl_2$ /0. 2 mg/mL 3xFLAG peptide, followed by 2 CV of buffer D/1 mM DTT/2 mM calcium chloride/0. 1 mg/mL 3xFLAG peptide. The eluates were pooled and applied to 1 mL of calmodulin affinity resin (Agilent

Technologies). Beads were washed with 20 CV of buffer D/1 mM DTT/2 mM CaCl$_2$ and bound protein eluted with 8 CV of buffer D/1 mM DTT/2 mM EGTA/2 mM EDTA. The eluates from the calmodulin beads were further fractionated on a MiniQ PC 3.2/3 anion exchange column equilibrated in 25 mM Tris pH 7.2/10 % glycerol/0.005 % Tween 20/1 mM DTT/150 mM KCl. Bound protein was eluted with a 30 CV gradient from 150 to 1000 mM KCl in 25 mM Tris pH 7.2/10 % glycerol/0.005 % Tween 20/1 mM DTT. CMG-containing peak fractions were pooled, dialyzed against 2 L of 25 mM HEPES-KOH pH 7.6/200 mM potassium acetate/2 mM magnesium acetate/10 % glycerol/1 mM DTT, aliquoted, flash-frozen in liquid nitrogen, and stored at –80 °C.

## DNA replication assays

All replication reactions were performed on linearized template with or without R-loop; 4 nM template DNA was incubated with 5 U *NsiI* in an 8 µL reaction containing 50 mM KOAc, 25 mM Hepes-KOH pH 7.5, 5 % glycerol, 2.5 mM DTT, 0.02 % NP-40, and incubated for 30 min at 37°C.

The salt concentration was increased to 100 mM KOAc; 16 nM ORC, 50 nM Cdc6, 100 nM Cdt1·Mcm2-7, and 5 nM ATP were added to the reaction and incubation continued for 20 min at 30 °C; and 150 nM of DDK was added and incubation continued for 20 min at 30 °C. Subsequently, a mastermix was added containing 50 µg of BSA, 20 nM Sld3-7, 125 nM Cdc45, 80 nM each dNTP, 16 nM CDK, 100 nM GINS, 20 nM Pol ε, 8 nM Dpb11, 30 nM Sld2, 150 nM RPA, 75 nM Pol α, 150 nM Ctf4, 25 nM RFC, 75 nM PCNA, 2 nM Pol δ, 0.2 mM each rNTP, 30 nM Top1, 20 nM Mrc1, 20 nM Csm3-Tof1. The salt concentration was increased to 180 mM KOAc and origin firing was induced by the addition of 14 nM Mcm10 in the presence of 5 µCi α-[32P]-dATP and incubation continued for 1 hr at 30 °C. Reactions were stopped by addition of 40 mM EDTA, 0.8 U Proteinase K, and 0.8 % SDS, followed by incubation for 30 min at 37 °C. Reactions were phenol/chloroform extracted and filtered through Illustra MicroSpin G-25 spin columns (GE).

For denaturing gels, 10 µL of each sample was digested with *ClaI* (Neb) in 1 × CutSmart buffer and subsequently quenched with 40 mM EDTA and loading dye. Digested samples were fractionated on 0.8 % agarose gels in 30 mM NaOH/2 mM EDTA. Denaturing gels were neutralized and fixed in 5 % TCA. For native gels, 10 µL of undigested sample were supplemented with loading dye and fractionated on 0.8 % native agarose gels in TAE. Denaturing and native gels were dried on Whatman paper and analyzed by phosphor imaging. Signal intensities were quantified using ImageJ.

For conventional time course experiments, aliquots were removed at desired times after the addition of Mcm10 and reactions stopped with 40 mM EDTA. For pulse-chase experiments, the concentration of dATP was reduced to 2 nM. Reactions were pulsed for 2.5 min by adding 5 µCi of α-[32P]-dATP along with Mcm10 and chased by addition of 1 mM cold dATP. Aliquots were removed and quenched with 40 mM EDTA at desired time points.

Unless indicated otherwise, Pif1 (0.7 nM), RNase H1 (1 nM), PDS (0.5 µM; Sigma), Fen1 (15 nM), and Cdc9 (15 nM) were added to the reactions alongside Mcm10. For treatment of templates with T4 PNK, 5 U of T4 PNK (NEB) were included in the initial *NsiI* digestion reaction and incubated at 37 °C for 1 hr.

For two-dimensional gel analyses, a 20 µL standard replication reaction was digested with ClaI, fractionated in the first dimension by 0.85 % native agarose gel-electrophoresis, the sample lane excised from the native gel and fractionated on a denaturing agarose gel overnight with buffer circulation in 30 mM NaOH/2 mM EDTA. The gel was neutralized by shaking in 5 % TCA, dried on Whatman paper, and analyzed by phosphor imaging.

## Polymerase assay

Four nM template was incubated in a 16 µL reaction volume with 0.5 U T4 PNK and 5 U *NsiI* in 50 mM KOAc/25 mM Hepes-KOH pH 7.5/5 % glycerol/2.5 mM DTT/0.02 % NP-40 and incubated for 1 hr at 37°C. Subsequently, the reaction was supplemented with 5 nM ATP, 50 µg of BSA, 80 nM each dNTP, 150 nM RPA, 25 nM RFC, 75 nM PCNA, 0.2 mM each rNTP. Reactions were supplemented with 5 µCi α-[32P]-dATP, followed by addition of Pol ε (20 nM), Pol α (75 nM), and Pol δ (2 nM), as indicated, and incubated for 1 hr at 30 °C. Reactions were stopped by addition of 40 mM EDTA, 0.8 U Proteinase K, and 0.8 % SDS, followed by incubation for 30 min at 37 °C. Reactions were phenol/chloroform extracted and filtered through Illustra MicroSpin G-25 spin columns (GE); 10 µL of each reaction sample was fractionated by denaturing agarose gel-electrophoresis.

## CMG helicase assays

Assays were performed in helicase assay buffer (20 mM HEPES-KOH pH 7.6/100 mM potassium acetate/10 mM magnesium acetate/0.1 mg/mL BSA/2.5 mM DTT). For time course analyses, 25 µL reactions containing 4 nM templates and 20 nM CMG were supplemented with 100 µM ATPγS and incubated at 30 °C for 1 hr. Unwinding was initiated by addition of 25 µL of helicase assay buffer/10 mM ATP to the reaction mixture. Incubation was continued at 30 °C and 6 µL aliquots were withdrawn at indicated time points. The reactions were stopped by adding 20 mM EDTA and 0.12 % SDS. Reaction products were fractionated by 10 % native PAGE in a Biorad Mini-PROTEAN system at 75 V for 2 hr in 0.5× TAE buffer. Gels were dried on a Whatman paper, and exposed to phosphorimager screen, and scanned on a Typhoon 7000 imager (GE Healthcare). The images were quantified using ImageJ and plotted using GraphPad Prism software.

For assays in *Figure 4*, 5 µL reactions containing 4 nM template, 20 nM CMG, and 100 µM ATPγS were incubated for 1 hr at 30 °C. Unwinding was initiated by adding 5 µL of helicase assay buffer/10 mM ATP. Reaction was stopped after 15 min by adding 10 µL of 40 mM EDTA and 0.24 % SDS.

## Acknowledgements

This work was supported by a MSKCC Functional Genomics Initiative grant (DR), NIGMS grants R01-GM127428 and R01-GM107239 (DR), NIH grant ES031635 (JDG), and NIH/NCI Cancer Center Support Grant P30 CA008748. We thank Sujan Devbhandari for help with CMG helicase assays, Ademola Adegbemigun for help with protein purification, and Iestyn Whitehouse for helpful discussions.

## Additional information

### Funding

| Funder | Grant reference number | Author |
|---|---|---|
| National Institute of General Medical Sciences | R01-GM127428 | Dirk Remus |
| National Institute of General Medical Sciences | R01-GM107239 | Dirk Remus |
| National Institutes of Health | P30 CA008748 | Dirk Remus |
| National Institute of Environmental Health Sciences | ES031635 | Jack D Griffith |

The funders had no role in study design, data collection and interpretation, or the decision to submit the work for publication.

### Author contributions

Charanya Kumar, Conceptualization, Formal analysis, Investigation, Methodology, Validation, Writing - original draft; Sahil Batra, Jack D Griffith, Investigation, Methodology, Validation; Dirk Remus, Conceptualization, Formal analysis, Funding acquisition, Project administration, Supervision, Validation, Writing - original draft, Writing - review and editing

### Author ORCIDs

Charanya Kumar http://orcid.org/0000-0001-7678-9288
Sahil Batra http://orcid.org/0000-0003-4210-3991
Dirk Remus http://orcid.org/0000-0002-5155-181X

### Decision letter and Author response

Decision letter https://doi.org/10.7554/eLife.72286.sa1
Author response https://doi.org/10.7554/eLife.72286.sa2

## Additional files

### Supplementary files
• Transparent reporting form

### Data availability
All data generated or analysed during this study are included in the manuscript and supporting files. Source data files have been provided.

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
