## [Decision Letter]

**Acceptance summary:**

The paper by Kumar et al., have studied in detail the process of DNA replication when it encounters a pre-formed R-loop in the DNA template, on either the leading strand template or the lagging strand template. R-loops are known to cause genome instability and the current paper nicely addresses how the replication machinery handles the different R-loops. They demonstrate that there are significant differences between the replication machinery encountering an R-loop on the leading strand versus the lagging strand. The presence of G4 quartet sequences also affects the replication fork stalling and this also differs between two template strands. They show that the RNase H1 ribonuclease and the Pif1 helicase also have differential effects on replication through R-loops on the leading and lagging strands.

**Decision letter after peer review:**

Thank you for submitting your article "The interplay of RNA:DNA hybrid structure and G-quadruplexes determines the outcome of R-loop-replisome collisions" for consideration by *eLife*. Your article has been reviewed by 3 peer reviewers, including Bruce Stillman as Reviewing Editor and Reviewer #1, and the evaluation has been overseen by Jessica Tyler as the Senior Editor. The following individual involved in review of your submission has agreed to reveal their identity: Philippe Pasero (Reviewer #3).

Kumar et al., have studied in detail the process of DNA replication when it encounters a pre-formed R-loop in the DNA template, on either the leading strand template or the lagging strand template. R-loops are known to cause genome instability and the current paper nicely addresses how the replication machinery handles the different R-loops. They demonstrate that there are significant differences between the replication machinery encountering an R-loop on the leading strand versus the lagging strand. The presence of G4 quartet sequences also affects the replication fork stalling and this also differs between two template strands. They show that the RNase H1 ribonuclease and the Pif1 helicase also have differential effects on replication through R-loops on the leading and lagging strands.

The paper has an extensive biochemical analysis of replication fork stalling and replication re-start. The data are excellent and support the conclusions.

The reviews below indicate that the data strongly support the conclusions about the interaction

The reviewers have listed some suggestions for improvement of the manuscript that should be addressed.

A. Two reviewers ask for a statement in the discussion about the in vivo context of R loops and topology of the template. These should be added.

1. The experiments described in this manuscript are beautifully designed and executed. However, a limitation of the in vitro approach used by the authors is that it requires the linearization of the plasmid template and therefore does not consider the contribution of DNA topology. Although this does not diminish the value of the study's conclusions, it would be worth mentioning somewhere in the manuscript that DNA supercoiling directly affect the formation and stability of R-loops (see PMID: 32107311) and plays a central role in transcription-replication conflicts. Reinvestigating the impact of R-loop promoting sequences in the context of topologically closed plasmids is an interesting challenge for the future.

2. There is one major limitation to this study, which should be noted in the text. It is important for the reader to recognize that this is a 'naked" R-loop and that the presence of RNA polymerase and other transcription factors could change some outcomes of the study. While this does not diminish the value of this work, the authors' mechanistic work provides an important advance to the field in understand fork progression at these structures upon which others are likely to build. However, adding this caveat to the discussion is suggested.

B. There are some clarifications of the paper that are needed and these are also listed below.*Reviewer #1 (Recommendations for the authors):*

Kumar et al., have studied in detail the process of DNA replication when it encounters a pre-formed R-loop in the DNA template, on either the leading strand template or the lagging strand template. R-loops are known to cause genome instability and the current paper nicely addresses how the replication machinery handles the different R-loops. They demonstrate that there are significant differences between the replication machinery encountering an R-loop on the leading strand versus the lagging strand. The presence of G4 quartet sequences also affect the replication fork stalling and this also differs between two template strands. They show that the RNase H1 ribonuclease and the Pif1 helicase also have differential effects on replication through R-loops on the leading and lagging strands.

The paper have an extensive biochemical analysis of replication fork stalling and replication re-start. The data are excellent and support the conclusions.

*Reviewer #2 (Recommendations for the authors):*

In this manuscript, the authors set out to determine the mechanism by which R-loops impede DNA replication, using a reconstituted yeast replication system and purified R-loops. They use a synthetic R-loop forming sequence which is known to contain a number of G-quadruplex forming sequences, and they generate purified R-loops with this sequence, adding them to their yeast extract system. In this way, and using the enzymes RNaseH or Pif1 alone or in combination to resolve the R-loop or the G-quadruplex respectively, the authors elegantly dissect the specific impacts of R-loops and G-quadruplexes on fork progression in an orientation-specific manner. They find that both R-loops and G-quadruplexes can affect fork progression in both the head-on and co-directional orientations, observing both fork stalling, uncoupling of DNA synthesis from replisome progression, nascent strand gaps and fork restart and clearly determining which element of the R-loop/G-guad is contributing to the observed effect. They find that the replisome can bypass R-loops that are annealed at the 5'end (i.e no tail) and unwind those with an unannealed/free 5'-end. They also find that the replisome can reprime DNA synthesis downstream of an RNA-DNA hybrid, using the RNA of the hybrid to prime restart. In the case of a G-quadruplex, they also observe efficient replication restart.

The authors did an excellent job in dissecting the different contributions of R-loops and G-quadruplexes on the outcome of replisome collisions, producing clean, high quality data that is well controlled. As a result, they were able to make several conclusions, all of which are strongly supported by their in vitro data. Their work also helps to explain several observations made in cells, where it can be hard to assess direct vs indirect impacts of the R-loop or the orientation of a collision and it will therefore be of significant value to the field. I do think there is one major limitation to this study, which should be noted in the text. I think it is important for the reader to recognize that this is a 'naked" R-loop and that the presence of RNA polymerase and other transcription factors could change some outcomes of the study. I want to emphasize that this does not diminish the value of this work, and think that the authors' mechanistic work provides an important advance to the field in understand fork progression at these structures upon which others are likely to build.

P3, line 15-17, Please add Pasero's reference on the fact that only some R-loops lead to DNA damage (PMID: 32769985)

P7, lines 12, 13 – please further explain the conclusion that leading strand stalling at CD and HO R-loops results from both fork stalling and helicase uncoupling in reference to the 2D gel analysis being discussed. The link to these data is not obvious to the reader.

Figure 3, Figure supplement 2 – The discussion regarding the stability of G4 structures is a little difficult to follow (bottom of page 9, top of page 10). Why are these G quads stable in one orientation but not the other in ssDNA when the hybrids is removed by RNaseH. I think these points could be made clearer in the manuscript and are a bit confusing at this point. Are they suggested there are multiple G4s on that strand, some dependent on the hybrid? Perhaps adding this into the figure would help if that is the idea.

p.11, lines 11 and 12 The authors talk about the G4 potential being less on the C-rich strand. At this point in the manuscript, it is not clear how they rule out the possibility that it is a G4 on the G rich strand that is actually affecting stalling. Later on p.11 (lines 17-16-18), the authors really test the idea that it is the G4 on the C-rich strand that affects fork stalling on CD strand by deleting a specific G4 forming sequence. I thus think that this and a few of their earlier conclusions/statements about the location of the problematic G4 are a bit premature and confusing to the reader. Unless I misunderstand this, I suggest they reorganize and/or rewrite this part of the manuscript to make this clearer as this conclusion is only really appropriate upon completion of this deletion experiment.

*Reviewer # 3 Recommendations for the authors:*

It is now well established that R-loops and G-quadruplexes (G4) are intrinsically difficult to replicate but the mechanism(s) by which they interfere with DNA replication has remained poorly understood in eukaryotes. In this manuscript, Kumar and colleagues used a well-characterized reconstituted DNA replication system to investigate how budding yeast replisomes deal with R-loops depending on their relative orientation and on the presence of accessory factors (RNase H1, Pif1, Tof1…). This smart experimental setup allowed them to detect fork stalling, helicase uncoupling and repriming with an unprecedented resolution and accuracy.

This study is important because it provides a wealth of novel information regarding the impact of R-loops on DNA synthesis. For instance, its shows that R-loops arrest forks in a Tof1-independent manner and act therefore differently than proteinaceous replication fork barriers (RFBs) such as tDNA and rDNA RFBs. Moreover, this work reveals that R-loops can interfere with DNA synthesis on both strands and in both orientations (head-on and co-directional), even though the CMG replicative helicase is able to unwind RNA:DNA hybrids and DNA duplexes with similar efficiencies. They can also induce gaps on both the leading strand and the lagging strand and promote repriming by Pol α. In addition, the authors provide direct evidence that R-loops promote G-quadruplex formation on the displaced DNA strand and that G4s can interfere with DNA synthesis in an orientation-dependent manner even after degradation of RNA:DNA hybrids with RNase H1. Together with other important findings reported in the manuscript, these observations shed new light on the complex interactions between the replisome, RNA:DNA hybrids and G4 structures, and challenge our view of transcription-replication conflicts. Overall, the data are of high quality and the manuscript is very well written. However, important issues need to be addressed prior to publication.

1. The experiments described in this manuscript are beautifully designed and executed. However, a limitation of the in vitro approach used by the authors is that it requires the linearization of the plasmid template and therefore does not consider the contribution of DNA topology. Although this does not diminish the value of the study's conclusions, it would be worth mentioning somewhere in the manuscript that DNA supercoiling directly affect the formation and stability of R-loops (see PMID: 32107311) and plays a central role in transcription-replication conflicts. Reinvestigating the impact of R-loop promoting sequences in the context of topologically closed plasmids is an interesting challenge for the future.

2. The EM and AFM images of R-loop structures displayed in Figure 1 and S1 are very nice, but the AFM images differ significantly from the ones recently published by the Vanoostuyse lab (Carrasco-Salas et al., 2019). Was formamide used for AFM experiments as well?

3. The authors have successfully used the S9.6 monoclonal antibody to detect RNA:DNA hybrids by EM but the number of observed structures is not indicated. How reliable is this observation? Could it be applied to detect G-quadruplexes by EM, for instance with the BG4 antibody?

4. The time course experiments shown in Figure 2 —figure supplement 2 are very important and should be shown in the main figure, at least the graphs.

5. Page 9, the authors discuss evidence that CD R-loops on the leading strand induce fork stalling but are dispensable for restart. Since restart depends on transcription but is not sensitive to RNase H1 treatment, their data suggest that G4 are involved. How could a G4 structure on the lagging strand promote repriming on the leading strand?

6. As reported by the Luke lab (Lockhart et al., 2019), RNase H1 and H2 are differentially required during the cell cycle. Moreover, RNase H2 interacts physically with PCNA and could play a different role at stalled replisomes, especially in the resolution of blocked lagging strand synthesis (page 19). Since the function of RNase H2 was tested here, this limitation should be discussed.

7. Page 5, line 11: ARS305 is a genetic element and should be italicized. Restriction enzymes (ClaI, NsiI) should be italicized as well.

---

## [Author Response]

Kumar et al., have studied in detail the process of DNA replication when it encounters a pre-formed R-loop in the DNA template, on either the leading strand template or the lagging strand template. R-loops are known to cause genome instability and the current paper nicely addresses how the replication machinery handles the different R-loops. They demonstrate that there are significant differences between the replication machinery encountering an R-loop on the leading strand versus the lagging strand. The presence of G4 quartet sequences also affects the replication fork stalling and this also differs between two template strands. They show that the RNase H1 ribonuclease and the Pif1 helicase also have differential effects on replication through R-loops on the leading and lagging strands.The paper has an extensive biochemical analysis of replication fork stalling and replication re-start. The data are excellent and support the conclusions.The reviews below indicate that the data strongly support the conclusions about the interactionThe reviewers have listed some suggestions for improvement of the manuscript that should be addressed.

We thank the reviewers for the constructive comments.

A. Two reviewers ask for a statement in the discussion about the in vivo context of R loops and topology of the template. These should be added.1. The experiments described in this manuscript are beautifully designed and executed. However, a limitation of the in vitro approach used by the authors is that it requires the linearization of the plasmid template and therefore does not consider the contribution of DNA topology. Although this does not diminish the value of the study's conclusions, it would be worth mentioning somewhere in the manuscript that DNA supercoiling directly affect the formation and stability of R-loops (see PMID: 32107311) and plays a central role in transcription-replication conflicts. Reinvestigating the impact of R-loop promoting sequences in the context of topologically closed plasmids is an interesting challenge for the future.

We thank the reviewer for pointing out this oversight and have added the following sentence at the end of the discussion: “Moreover, since DNA supercoiling affects R-loop formation and stability (Chedin and Benham, 2020), it will be of interest to investigate the impact of R-loops on replication fork progression in topologically closed DNA templates.”

2. There is one major limitation to this study, which should be noted in the text. It is important for the reader to recognize that this is a 'naked" R-loop and that the presence of RNA polymerase and other transcription factors could change some outcomes of the study. While this does not diminish the value of this work, the authors' mechanistic work provides an important advance to the field in understand fork progression at these structures upon which others are likely to build. However, adding this caveat to the discussion is suggested.

As explained in the introduction (page 4, lanes 16-22), the purpose of our study was to exclude additional components such as RNAP, transcription factors, and chromatin that are normally present at R-loop-containing regions in order to specifically assess the impact of the R-loop structure itself on replisome progression. However, in order to clarify this point as suggested we have now added the following statement to the last paragraph of the discussion: “While the experiments reported here were designed to specifically assess the impact of the R-loop nucleic acid structure on replisome progression, many additional factors, such as RNAP, transcription factors, and specialized chromatin structures are associated with R-loops in vivo and further influence the outcome of R-loop-replisome collisions (Castellano-Pozo et al., 2013; Garcia-Muse and Aguilera, 2019; Garcia-Pichardo et al., 2017; Garcia-Rubio et al., 2018). Reconstitution of R-loops in chromatin templates will thus be an interesting challenge for future studies.”